# The 3D architecture of the pepper genome and its relationship to function and evolution

Yi Liao[1,2,10], Juntao Wang [1,3,10], Zhangsheng Zhu[1,3], Yuanlong Liu[4,5,6], Jinfeng Chen[7], Yongfeng Zhou [8], Feng Liu[9], Jianjun Lei[1,3], Brandon S. Gaut [2], Bihao Cao [1,3✉], J. J. Emerson [2✉] & Changming Chen [1,3✉]

The organization of chromatin into self-interacting domains is universal among eukaryotic genomes, though how and why they form varies considerably. Here we report a chromosome-scale reference genome assembly of pepper (*Capsicum annuum*) and explore its 3D organization through integrating high-resolution Hi-C maps with epigenomic, transcriptomic, and genetic variation data. Chromatin folding domains in pepper are as prominent as TADs in mammals but exhibit unique characteristics. They tend to coincide with heterochromatic regions enriched with retrotransposons and are frequently embedded in loops, which may correlate with transcription factories. Their boundaries are hotspots for chromosome rearrangements but are otherwise depleted for genetic variation. While chromatin conformation broadly affects transcription variance, it does not predict differential gene expression between tissues. Our results suggest that pepper genome organization is explained by a model of heterochromatin-driven folding promoted by transcription factories and that such spatial architecture is under structural and functional constraints.

[1] Key Laboratory of Biology and Genetic Improvement of Horticultural Crops (South China), Ministry of Agriculture and Rural Affairs, College of Horticulture, South China Agricultural University, Guangzhou 510642, China. [2] Department of Ecology and Evolutionary Biology, University of California, Irvine, CA 92697, USA. [3] Lingnan Guangdong Laboratory of Modern Agriculture, Guangzhou 510642, China. [4] Department of Computational Biology, University of Lausanne, Lausanne, Switzerland. [5] Swiss Cancer Center Leman, Lausanne, Switzerland. [6] Swiss Institute of Bioinformatics, Lausanne, Switzerland. [7] State Key Laboratory of Integrated Management of Pest Insects and Rodents, Institute of Zoology, Chinese Academy of Sciences, Beijing 100101, China. [8] Shenzhen Branch, Guangdong Laboratory of Lingnan Modern Agriculture, Genome Analysis Laboratory of the Ministry of Agriculture and Rural Affairs, Agricultural Genomics Institute at Shenzhen, Chinese Academy of Agricultural Sciences, Shenzhen 518120, China. [9] College of Horticulture, Hunan Agricultural University, Changsha 410128, China. [10] These authors contributed equally: Yi Liao, Juntao Wang. ✉email: caobh01@scau.edu.cn; jje@uci.edu; cmchen@scau.edu.cn

The folding of chromosomes into self-interaction domains[1], also known as topologically associating domains (TADs), appears to be conserved in evolution[2]. TADs and similar structures occur in diverse groups of eukaryotes, from fungi and bacteria to plants and animals[3]. Many mechanisms have been proposed for their formation, of which loop extrusion and compartmentalization are two leading models in animal systems[4–7]. While evidence suggests that these mechanisms may operate in tandem to jointly establish or maintain the spatial organization of the genome, the prevalence of each differs across species[8–11]. Like animals, TAD-like domains have been observed from Hi-C analyses of many plants; however, the mechanisms by which they form (and whether they are shared with animals) are largely unknown[2,12]. Additionally, TADs organized by different mechanisms may exhibit distinct structural and functional properties[8,13–15]. Thus, clarifying the formation mechanisms of TADs is necessary for further elucidating their functional specialization.

Unlike in animals, where TADs can be readily detected genome-wide, small plant genomes like *Arabidopsis thaliana* and its close relative *Arabidopsis lyrata* carry few such domains[16]. However, other plant species with relatively large genome sizes do exhibit more pronounced chromatin domain architectures[17–20]. Comparisons between plant species imply that TAD prevalence in plants may be associated with genome size or other sequence properties, like the linear distribution of genes, regulatory elements, and transposable elements[12,21,22]. Consequently, 3D genome organization appears to exhibit great diversity in plants. This may also be true of the mechanisms that contribute to TAD-like folding domain formation. For example, TAD-like domains in maize and tomato are reported to largely coincide with compartments, suggesting their formation is associated with compartmentalization in these species[18]. Recent studies in wheat[19] have reported that a large proportion of chromatin domains are demarcated by gene-to-gene loops, and the genome is organized into regions of relatively high transcription-i.e. transcription factories[23]. Many other features such as transcription factors are also found to be associated with the formation of plant chromatin domains[14,17]. Thus, in plants, there appears to be variation not only in the prevalence of topological domains but also in their mechanism of formation.

TADs are thought to behave as functional and structural units of the genome in evolution[5]. In metazoans, chromosomal rearrangement breakpoints rarely occur within TAD bodies, implying that disruption of TAD integrity is unfavorable and subject to purifying selection[24–28]. Chromatin structures are also found to be associated with patterns of both somatic mutation[29] and genomic variants across evolutionary timescales[30]. Furthermore, long-range promoter-enhancer contacts that form loops are known to constrain large-scale genome evolution[31]. Given that the spatial organization of the genome affects organismal function, an open question in plant biology is: how does natural selection affect the acquisition and fate of mutations—particularly, structural variants—that alter spatial organization? In plants, even though 3D genome organization is thought to play an important role in the polyploidization process[32–35], our understanding of the relationship between chromatin architecture and structural variants remains incomplete.

Spatial genome organization is strongly associated with transcription. Numerous studies at the organismal[31,36], tissue[24], and cell type[37–39] levels have established that rearrangement of 3D chromatin organization (i.e. higher-order chromatin structures, such as loops, TADs, and compartments) is associated with changes in gene expression. However, many studies suggest that chromatin conformation is not required for *cis*-regulatory interactions that activate normal gene expression[40–42], and instead it

may primarily act as an architectural framework to facilitate gene regulation[43]. Although many recent attempts have been made to study these phenomena in plants[33,44–46], the relationship between 3D genome organization and the regulation of transcription in plant systems remains elusive.

In this work, we investigate 3D genome organization and its functional implications by integrating a new de novo chromosome-scale long-read genome assembly with Hi-C, epigenomic, transcriptomic, and genetic variation data in pepper (*C. annuum*). We choose this species both because of its extensive cultivation and because its 3D chromatin architecture exhibits clear interaction domains that span most of the genome, comparable to observations in *Drosophila* and mammals (Supplementary Fig. 1). Our results suggest that chromatin architecture in pepper is characterized by heterochromatin-driven domains, which are likely sculpted by transcription factories. We use genetic variation data to show that domain organization is likely under structural constraints with functional consequences. Our results expand our understanding of the mechanistic and functional principles of chromosome folding in plant genomes.

## Results

**A chromosome-scale genome assembly of *C. annuum*.** We chose to sequence the pepper (*C. annuum*) inbred line CA59 (Supplementary Fig. 2) for its desirable agronomic characteristics[47]. We performed de novo assembly of the genome using ~415.9 Gb Pacific Biosciences (PacBio) long-read sequence data (153× genomic coverage), ~362.0 Gb (123×) short-read sequence data (150 bp paired-end, BGI genomics), and ~415.2 Gb (141×) Hi-C data (150 bp paired-end, BGI genomics) (Supplementary Table 1 and Supplementary Fig. 3). Assembling of PacBio long reads alone produced a draft assembly that had 633 gapless contigs with a contig N50 of 41.3 Mb (Supplementary Table 2). Such high continuity is likely a consequence of low heterozygosity (0.23%) in our sample and the length of reads (subread N50 was 28,351 bp). The draft assembly was polished with short reads until reaching an estimated Phred quality score of QV52 (see Methods). Using Hi-C linkage information, 505 out of the 633 initial contigs were scaffolded into 12 pseudomolecules (scaffold N50 is 262 Mb) spanning 3.07 Gb sequences, leaving 128 unplaced contigs occupying only 11.66 Mb sequences (Supplementary Table 2). Our chromosome-scale assembly showed high collinearity with the previous Zunla-1 assembly[48] (Supplementary Fig. 4a) whose contigs were ordered and oriented via a high-density genetic map, providing corroborating evidence for the accuracy of the Hi-C scaffolding result. The total genome size of the final assembly was similar to the estimated value (~2.95 Gb) based on a k-mer frequency analysis (Supplementary Fig. 5) and previous studies of pepper accessions[48–50].

Our chromosome-scale assembly recovers 95.8% of BUSCO (Benchmarking Universal Single-Copy Ortholog) genes (Embryophyta odb9 dataset), exceeding all previous *Capsicum* genome assemblies that were based on only Illumina sequencing (Supplementary Table 3). Moreover, de novo annotation of long terminal repeat retrotransposons (LTR-RTs) identified between 2917 and 4285 more full-length elements in our assembly than for previous assemblies (Supplementary Table 4), a likely consequence of higher continuity and completeness of our assembly in intergenic regions. Our assembly represents the first reference-quality genome assembly for pepper exceeding the EBP 6.C.Q40 standard[51].

Gene annotation was conducted by combining evidence from PacBio full-length mRNA sequencing data (Iso-Seq) generated from five tissues (leaf, bud, pulp, placenta, and root), protein sequences previously annotated in closely related genomes, and

ab initio prediction (Supplementary Table 5). A total of 46,160 protein-coding genes were predicted, which were enriched towards the ends of the chromosomes (Supplementary Fig. 4b), resembling observations in other large plant genomes. Preservation of synteny between genomes of pepper and three distantly related solanaceous species (tomato, eggplant, and potato) was thus common at chromosome ends (Supplementary Fig. 4c). We also annotated repeat content. Approximately 84.71% of the pepper genome was annotated as repetitive sequences, of which LTR-RTs alone make up 73.21% (Supplementary Table 6), including 59.89 Mb (1.95%) that represent 7,074 full-length elements (Supplementary Fig. 6a). This result suggests that the vast majority of LTR-RTs in the pepper genome are fragmented. Amongst annotated LTR-RTs, 7 families were abundant, with 50 or more copies in the genome per family, representing ~2,430 total insertions. Interestingly, most insertions in each of these seven families had identical 5′ and 3′ long terminal repeats, indicating recent bursts of retroposition. Additional structural analysis of LTR-RT elements along the chromosomes suggests illegitimate recombination is the major process driving the rapid decay of LTR-RTs in the pepper genome (Supplementary Fig. 6; see Supplementary Note 1 for more details).

**Hi-C interaction maps from four tissues**. To interrogate the 3D genome architecture of *C. annuum*, we generated in situ Hi-C data from four tissues including leaf, bud, pulp, and placenta, each with two biological replicates. A total of 5.54 billion raw Hi-C read pairs (2 × 150 bp) were produced, ranging from 557 to 788 million reads across samples, corresponding to raw sequencing coverages from 54x to 77x (Supplementary Table 7). We constructed Hi-C maps using both HiCExplorer[11] (Supplementary Table 8) and Juicer[52] (Supplementary Table 9). All Hi-C maps achieved a resolution around or higher than 10 kb (Supplementary Table 10), following previously described methods[8]. Quality assessment using 3DChromatin_ReplicateQC toolkit[53] shows that our Hi-C data are of high quality as evidenced by QuASAR quality scores (0.039–0.061)[54] (Supplementary Table 11) and agreements between replicates (Supplementary Table 12). The reproducibility of Hi-C maps between biological replicates was also supported by the Pearson correlation analysis of their contact frequencies (Supplementary Fig. 7a).

Inspection of the Hi-C maps revealed that the contact density was strongly concentrated along the main diagonals (Fig. 1a and see Juicer Hi-C maps in Supplementary Fig. 7b), suggesting 3D proximity of pairs of loci is highly correlated with their linear genomic distance, as expected. As in other large plant genomes[18], we observed an X-shaped trans-interaction pattern, though we observe it only in certain tissues, like leaf and bud, but not in pulp and placenta (Fig. 1a and Supplementary Fig. 7b). The anti-diagonal pattern has been suggested as reflective of the chromosome "Rabl" configuration within the nucleus[55,56]. We also observed significantly more (*P* < 0.0005) long-range (>20 Mb) versus short-range contacts (<20 Mb) in leaf and bud compared to pulp and placenta (Fig. 1b–d and Supplementary Fig. 7c–e). These results demonstrate how global chromosome conformation inside nuclei might differ between cells from different plant tissues.

**Subcompartment patterning is associated with genomic and epigenomic profiles**. A PCA-based analysis (see Methods) of 500-kb resolution Hi-C contact data segmented the pepper chromosomes into clearly defined "A" and "B" compartments. As with observations in other large plant genomes, "A" compartments were concentrated near telomeres whereas "B" compartments occupied the large middle repetitive regions of

chromosomes (Fig. 2a), corresponding to the global distribution of gene and TE sequences. However, because PCA approaches failed to recover segments with consistent biological properties at higher resolutions, we applied a hierarchical approach called Calder[13] that iteratively bisects the genome into nested sub-compartments (i.e., first into two compartments; then from two to four and from four to eight). When applied to higher-resolution matrices (10-kb and 40-kb bins), we inferred sub-compartments with mean lengths of ~250-kb and 300-kb, respectively (cf. Fig. 2b and Supplementary Fig. 8a). Although subcompartments identified at both resolutions were globally consistent with each other, the 10-kb matrices assigned more genome regions to B subcompartments than the 40-kb matrices (i.e., 59–65% vs. 48–55%) (Fig. 2c and Supplementary Fig. 8b), suggesting higher-resolution Hi-C matrices permit compartment classification on a finer scale.

To evaluate biological information captured at different levels of subdivisions (i.e., 4 and 8 subcompartments), we measured the association of subcompartments with genomic and epigenomic features. We measured DNA methylation and histone modification with ChIP-seq of young leaf tissue (e.g., H3K4me3, H3K27me3, and H3K9me2). Subcompartment rank (decreasing from A1 to B2 for four subcompartments or from A1.1 to B2.2 for eight subcompartments) is strongly associated with these genomic and epigenomic features (see results for subcompartments annotated at 10-kb resolution in Fig. 2d, e and Supplementary Fig. 9 and for 40-kb in Supplementary Fig. 8c). "A" subcompartments are generally enriched for genes, CHH DNA methylation, and ChIP-seq signals for H3K4me3 which tend to mark active chromatin states. Because "A" subcompartments are enriched for genes, they are also enriched for a repressive mark associated with genes, namely H3K27me3. In contrast, 'B' subcompartments are enriched for LTR retrotransposons, CpG and CHG DNA methylation, and ChIP-seq signals for H3K9me2, which tends to be enriched within repressed chromatin. Notably, the active "A" subcompartments exhibited elevated overall DNA methylation levels (i.e., summed across all three contexts) than the inactive "B" subcompartments, consistent with the previous results[57]. Overall DNA methylation levels are positively correlated with subcompartment rank with the exception of A2 (four subcompartments) and A2.2 (eight subcompartments), which exhibited lower methylation levels than other A subcompartments. These observations suggest that the inferred multi-scale subcompartments in the pepper genome may reflect domains with subtle differences in the epigenetic modifications and such differences may govern contact patterns.

We also evaluated the consistency of compartments across tissues, with 82–89% of the genome sharing the same major A/B compartment label across pairwise sample comparisons at 10-kb resolution (Fig. 2f). Though this drops to between 31 and 51% at the eight subcompartment level (Fig. 2f), more than 90% of subcompartments received the same label or the label within the two closest adjacent ranks (Fig. 2g), suggesting that compartments are often preserved across tissues. Similar results were obtained for subcompartments identified when using 40-kb matrices (Supplementary Fig. 8d, e).

**Chromatin interaction domains occupy a substantial portion of the pepper genome and are generally preserved across tissues**. We next used the Hi-C data to explore and annotate chromatin folding domains (i.e., TADs), using three programs (Arrowhead, HiCExplorer, and TopDom). Although we detected considerable variation in TAD calls, all of the approaches revealed the presence of clear TAD-like domains (Fig. 3a and Supplementary Fig. 10a). For example, using a 40-kb resolution leaf Hi-

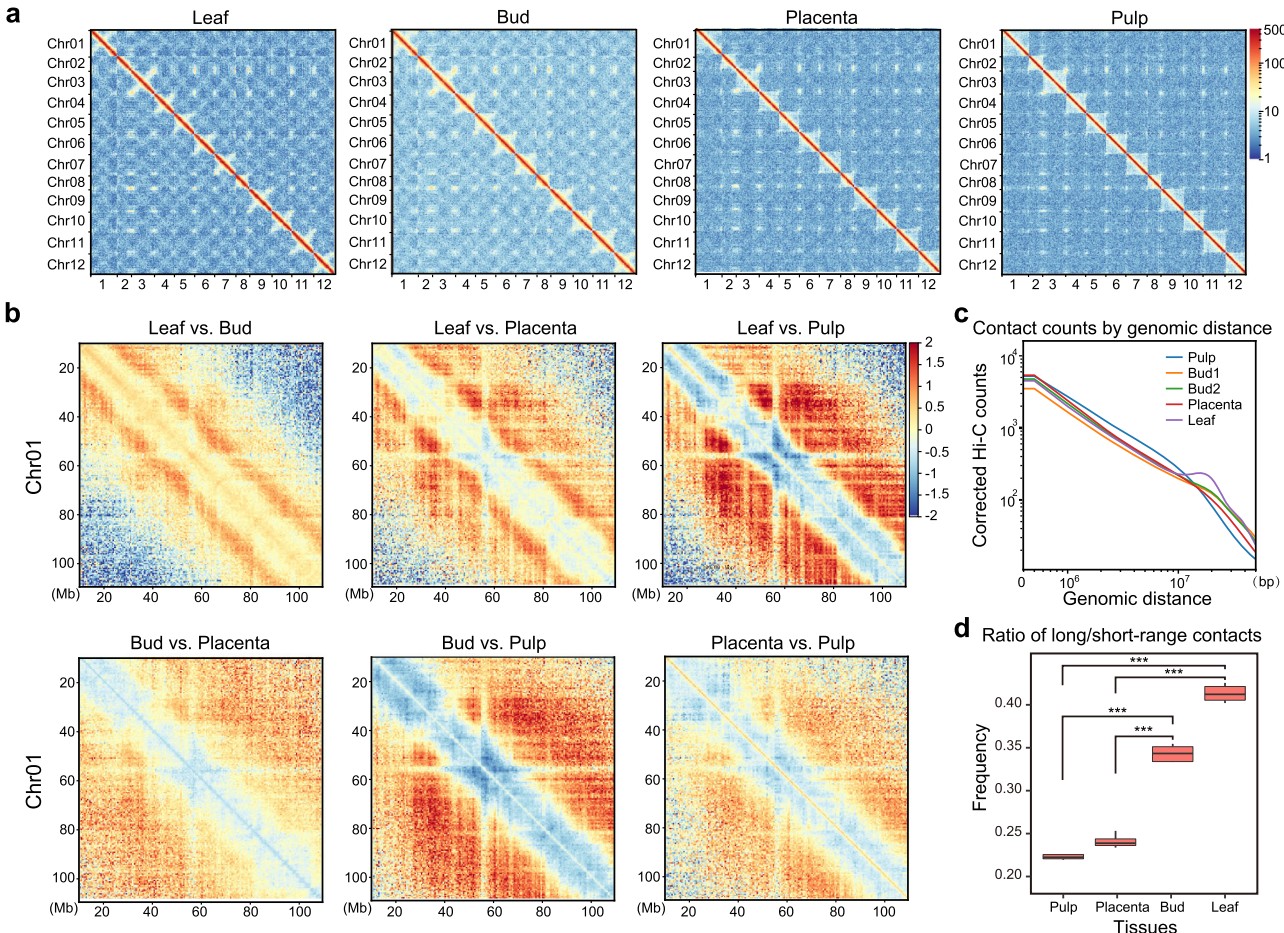

**Fig. 1 Hi-C interaction matrices generated from four tissues of *C. annuum*. a** Genome-wide normalized and corrected Hi-C maps (HiCExplorer) at 500 kb resolution. In the leaf and bud, an X-shaped trans-interaction signal appears within each chromosome, while it is weaker or not evident in contact maps of pulp and placenta. See also Juicer Hi-C maps in Supplementary Fig. 7b. **b** The $\log_2$-transformed ratio of Hi-C matrices between tissues. Red designates enrichment in the first tissue and blue depletion. **c** The genomic distance vs. contact counts plot using Hi-C matrices at 500 kb resolution. Leaf and Bud show enrichment of long-range contacts (>20 Mb) than pulp and placenta. Only samples in the first batch were shown. See samples in the second batch in Supplementary Fig. 7c and results based on Juicer Hi-C maps in Supplementary Fig. 7d. **d** The ratio of long-range (>20 Mb) versus short-range contacts. The sample size for the boxplot is the number of chromosomes ($n = 12$). This ratio is significantly higher in leaf and bud compared to pulp and placenta. The boxplot represents the median (band inside the box), first and third quartiles. Whiskers extend to 1.5 times the interquartile range (IQR). $P$ values (***$p < 0.0005$) were derived from two-side Wilcoxon matched-pairs signed-rank tests. For results from juicer Hi-C maps, see Supplementary Fig. 7e. Source Data underlying Fig. 1d is provided as a Source Data file.

C map, these three methods identified 1680, 4663, and 2641 domains, with medium sizes of 1180, 651, and 1152 kb, and occupying ~55, 99, and 99% of the genome, respectively (Supplementary Fig. 10b, c). Even so, a substantial number of TAD-like domains (1911) were consistently identified by at least two methods (Supplementary Fig. 10c), comparable to what we previously observed in *Drosophila*[28], and these TAD-like domains covered 55.4% of the genome. As in animals[58], they are organized in a hierarchical fashion, such that small domains often reside within larger ones (Fig. 3b). Application of TADtool[59] (which is based on the algorithm, insulation index) to call TADs revealed that ~75% of the genome is covered by TAD-like domains, confirming results obtained with other approaches that chromatin folded into self-interacting domains is a prominent feature of genome architecture in pepper (Supplementary Note 2 and Supplementary Figs. 11, 12).

By analyzing domains inferred by TopDom, which performs well in TAD annotation in benchmark comparisons[60], we found that our domain calls were consistent across tissues both in location (Fig. 3c) and size (Fig. 3d). A hierarchical clustering analysis also demonstrated that domain calls were reproducible

across tissues and replicates (Fig. 3e). Roughly, between 58 and 79% of TAD-like domains (measured in their genome coverage), and between 60 and 91% of the boundaries were shared across pairwise sample comparisons (Supplementary Fig. 10d). At least 85% of domains identified in one tissue were also detected in other tissues (Fig. 3f). Of the domains found only in a single tissue, about 56.4–86.4% are found only in a single replicate, whereas 13.6–43.6% (which corresponds to 0.6–3.5% of the total domains) are found in both replicates. Similar results were obtained using TAD-like domains inferred from TADtool (Supplementary Note 2 and Supplementary Figs. 11, 12). Our results suggest that only a small fraction of domains might be limited to only one of the tissues investigated here. Future work with higher replication will permit rigorous annotation of tissue-specific domains, allowing us to quantify the degree of divergence and conservation between tissues.

**Characterization and classification of TAD-like domains.** To further characterize the TAD-like domains, we conducted hierarchical clustering of domains (TopDom calls based on the 40-kb

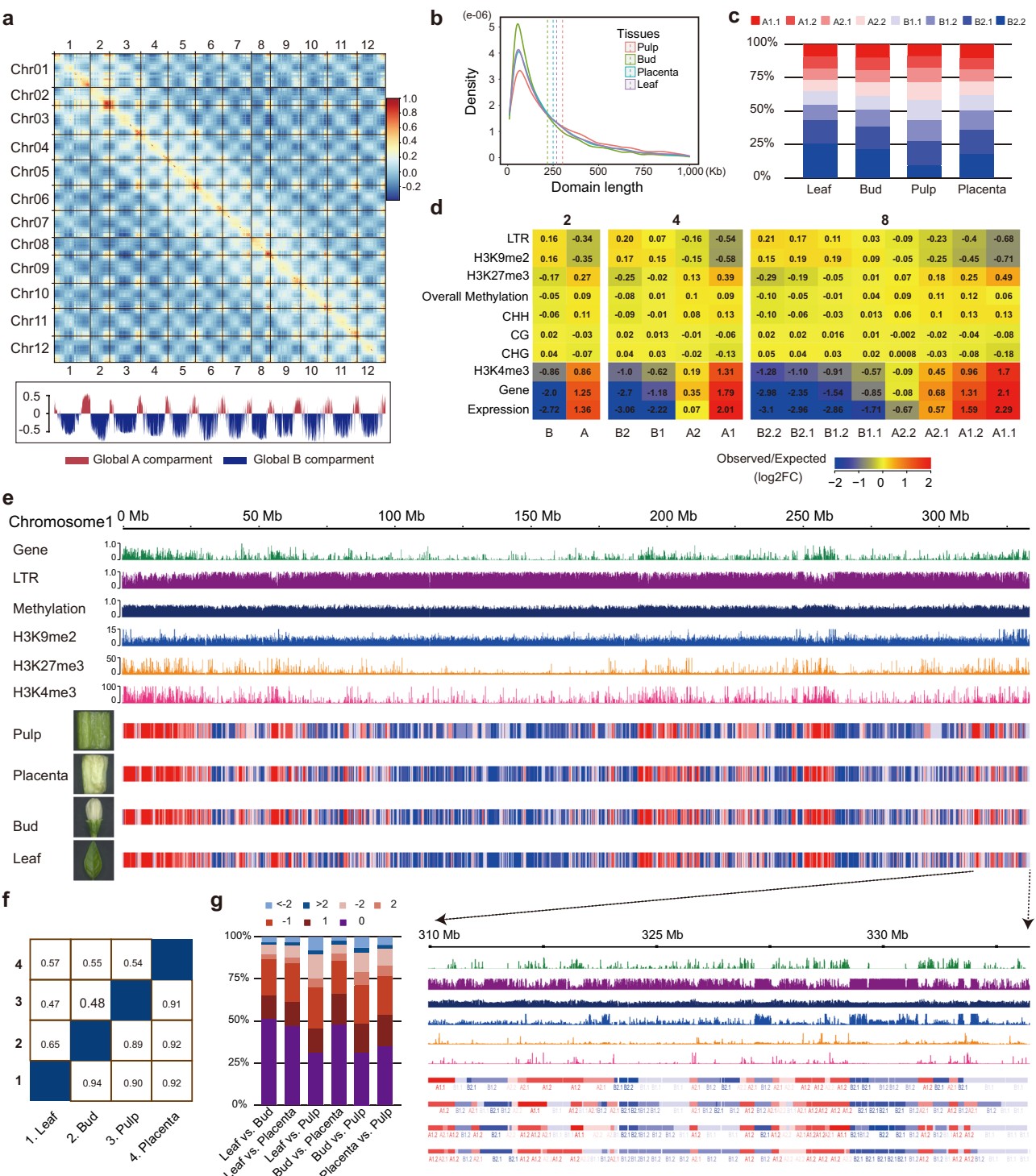

bin map of a leaf) based on a set of genomic (e.g., LTR and gene density) and epigenomic features (e.g., DNA methylation and histone modifications such as H3K4me3, H3K9me2, and H3K27me3) within their bodies and recovered three major groups (Fig. 4a). We labeled domains in group 1 ($n = 315$) as active because of their enrichment for genes and the active chromatin mark H3K4me3; domains in group 2 ($n = 1011$) as inactive, because of their high levels of DNA methylation sum across CG, CHG, and CHH contexts; and domains in group 3 ($n = 1315$) as HDF (heterochromatin-driven folding) because they are enriched for LTR retrotransposons and the hetero-chromatin mark H3K9me2 (Fig. 4b and Supplementary Fig. 13a).

HDF domains occupy ~60% of the genome (Supplementary Fig. 14a) and have a mean length of about 1.1 Mb, which is significantly larger than either active (734 kb) or inactive domains (698 kb) (Fig. 4c).

Approximately 89.2% of genomic regions in the active domains were assigned as "A" compartments, while 87.7% of genomic regions in the HDF domains were assigned as "B" compartments (Supplementary Fig. 14b), suggesting domains in the same group tend to belong to the same compartment. We found that ~36–45% of TAD-like domain boundaries called at 10-kb or 100-kb resolution overlapped with the boundaries of *Calder*-inferred subcompartments called at the same resolutions (Fig. 4d). Indeed,

**Fig. 2 Subcompartments are correlated with a number of genomic and epigenomic landscapes and maintained across tissues in the pepper genome.**
**a** Pearson correlation matrix heatmap (leaf 500-kb) shows the segregation of the pepper genome into global A/B compartments. The first principal component (PC1) derived from the analysis of this matrix was used to define the A and B compartments and is displayed below. Positive PC1 values are shown in red, representing A compartments, and negative PC1 values are shown in blue and designated as B compartments. **b** The size distribution of the *Calder*-inferred subcompartments using 10-kb resolution matrices across tissues. All samples display a roughly constant size distribution with a mean value of ~250 kb. **c** The genome-wide percentage of subcompartments called across tissues. Most (59–65%) of the genome is classified as inactive B subcompartments. **d** Enrichment analysis of genomic and epigenomic features (rows) across subcompartments (columns). Log2 fold changes between the observed median value and the expected median values are color-coded. Enrichment values were calculated based on Hi-C maps of 10-kb bin size. **e** Correspondence of subcompartments and the distribution of genomic (gene and LTR content) and epigenomic features (DNA methylation and histone modifications) shown for chromosome 1. Only DNA methylation level sum across all cytosine residues is shown. The tracks for individual tissues mark regions of A (red) or B (blue) compartments. A local example is shown below. **f** Similarity of the A/B compartments and subcompartments between tissues. The upper part of the matrix is shown for 8 subcompartments, while the lower part is shown for A/B compartments. **g** Subcompartment switching between tissues. Pairwise comparisons across four tissues were shown. Numbers above where "0" indicates unchanged subcompartment, "1", "2", and ">2" indicate subcompartment shift spanning 1, 2, or more than 2 subcompartments for lower ranks to higher ranks, and "−1", "−2", and "<−2" indicate subcompartment shift spanning 1, 2, or more than 2 subcompartments for higher ranks to lower ranks. For results based on 40-kb maps, see Supplementary Fig. 9. Source Data underlying Fig. 2b, c, g are provided as a Source Data file.

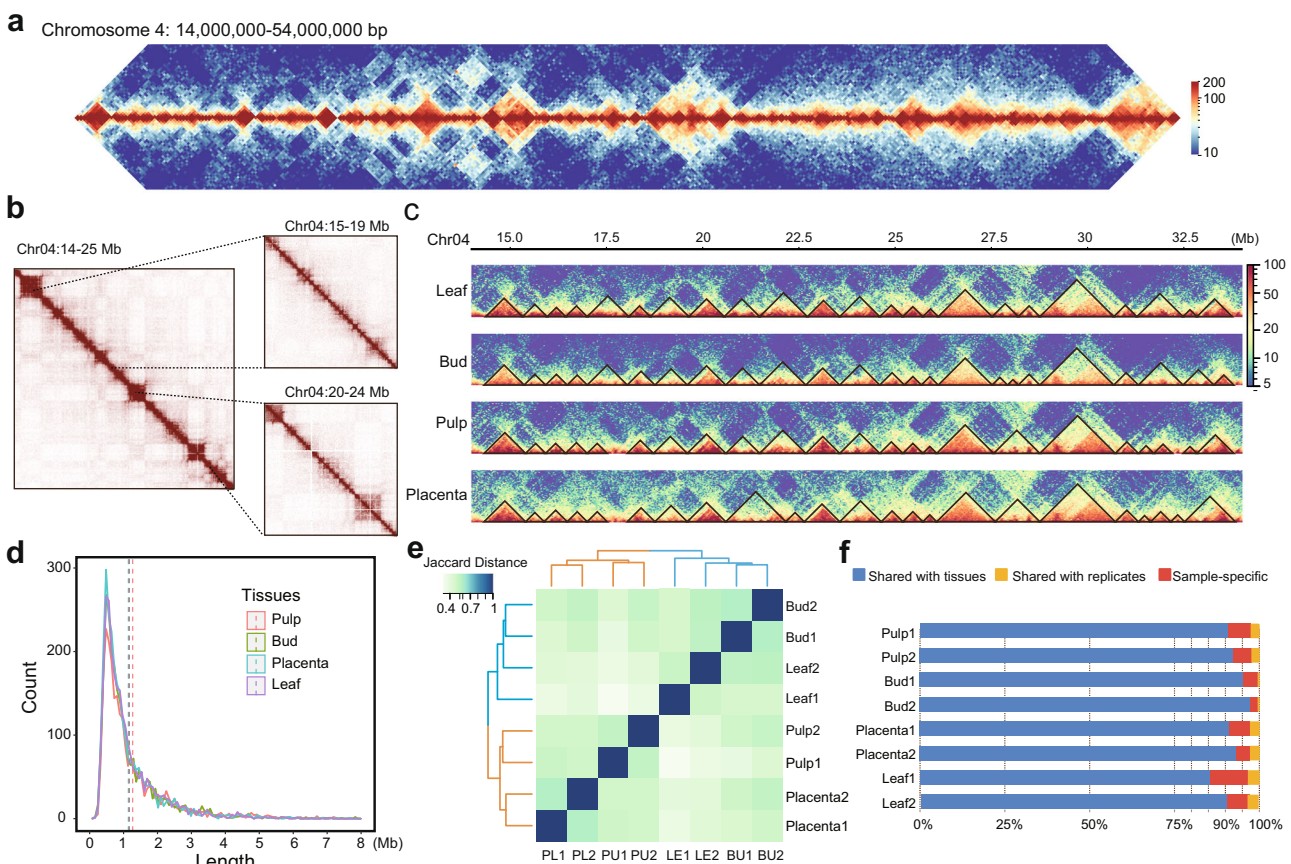

**Fig. 3 The pepper genome is organized into TAD-like folding domains. a** Example of TAD-like domains on a 40-Mb region of chromosome 4. This region is nearly completely segmented into TAD-like domains along their entire length. Leaf Hi-C interaction map at 100 kb resolution is shown. TADs are contiguous regions of enriched contact frequency that appear as squares in a Hi-C map. **b** Small contiguous domains show from Hi-C maps at higher resolutions, e.g., 40 kb (left), 10 kb (top right), and 5 kb (bottom right). **c** TAD-like domains are generally consistent across tissues. **d** The size distribution of TAD-like domains. Vertical dashed lines indicate mean values. **e** Hierarchical clustering analysis of TAD-like domains based on the Jaccard distance of their shared genome coverage across tissues and biological replicates. As expected, tissues are generally clustered together. **f** Conservation of domains across tissues. Domains used for analyses in (**c–f**) were annotated by TopDom at 40 kb resolution. Source Data underlying Fig. 3d–f are provided as a Source Data file.

at 10-kb resolution, ~18% of TAD-like domains called match perfectly with compartment domains, a fourfold enrichment over random expectation (domain body matches are defined by ≥80% reciprocal overlap, Supplementary Fig. 13b). We conclude that a large fraction of TAD-like domains in the pepper genome is compartment domains, consistent with previous findings[13].

As in animals, domain boundaries were enriched for genes and active chromatin marks (e.g., H3K4me3) but were depleted for inactive marks (e.g., H3K9me2) and LTR retrotransposons, a pattern reflected in all types of domain boundaries (Fig. 4e). Notably, the repressive histone mark H3K27me3 was neither enriched nor depleted at domain boundaries, contrasting with the

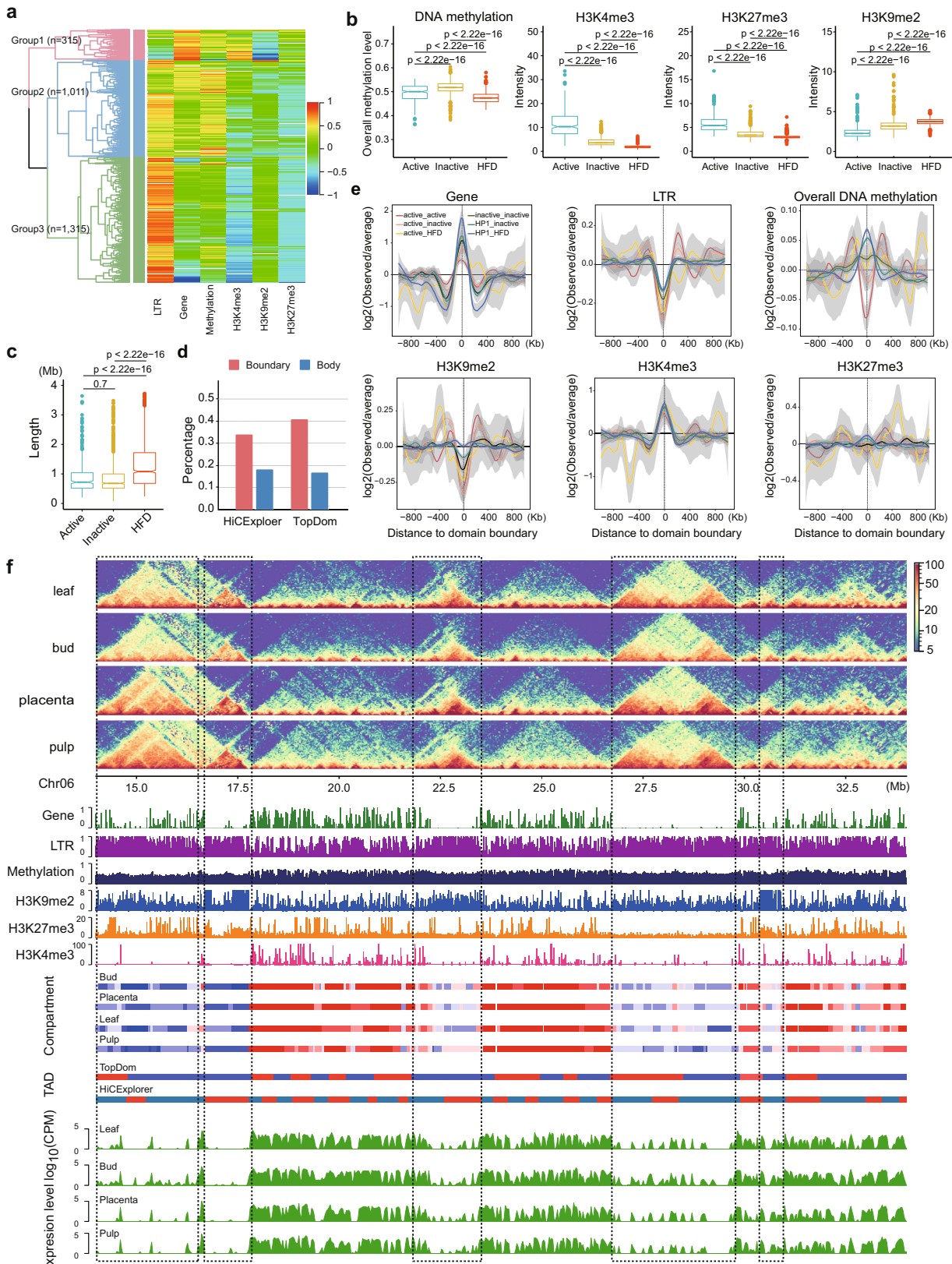

observation in *Drosophila* that shows depletion at TAD boundaries[28]. We also observed that domain boundaries exhibited lower CG and CHG methylation levels compared to their flanking regions but had higher CHH methylation (Supplementary Fig. 13c). Summed across all three methylation contexts, DNA methylation levels differed between boundary types. Generally, boundaries of active domains had lower levels of DNA methylation compared to their flanking regions, while boundaries of inactive and HDF domains had higher levels of DNA methylation (Fig. 4e).

Heterochromatin has recently been proposed as an important driving force of 3D genome folding of eukaryotic genomes[61–63].

**Fig. 4 Characterization and categories of TAD-like domains in the pepper genome. a** Hierarchical clustering analysis of TAD-like domains based on genomic and epigenomic features revealed three major groups. Domains were annotated by TopDom using the leaf 40-kb Hi-C map. **b** Domains between groups displayed significant differences in epigenomic features and were therefore classified as active ($n = 315$), inactive ($n = 1011$), and HDF ($n = 1315$). DNA methylation level was calculated based on the sum of sites across all contexts. See results for CpG, CHG, and CHH contexts, separately, in Supplementary Fig. 13a. All values were calculated from 10-kb bins. **c** Sizes vary across TAD groups. Domains in the HDF group are significantly larger than in the other two groups. Box plots in (**b**, **c**) represent the median (band inside the box), first and third quartiles. Whiskers extend to 1.5 times the IQR. Outliers were shown. $P$ values from two-sided Wilcoxon rank-sum tests. **d** Percentage of TAD-like domains and boundaries identified by HiCExplorer (red) and TopDom (blue) overlap with compartment domain and boundaries inferred by Calder using 10-kb resolution maps. For comparisons using 40-kb and 100-kb maps, see Supplementary Fig. 13b. **e** Genomic and epigenomic feature profiles centered at domain boundaries. Boundaries were classified into six groups based on their flanking domains. The standard error bounds were computed using the loess method based on a t-based approximation executed in ggplot's smooth geometry in R. **f** A representative example of folding domains in a 20-Mb region on chromosome 6. The below panels show genomic and epigenomic feature profiles, subcompartments inferred by Calder, TADs called by TopDom and HiCExplorer, and transcription profiles (measured in the 40-kb bin size) from four tissues. The black dashed rectangles highlight the heterochromatin folding domains which align with genomic regions enriched in retrotransposons and H3K9me2 mark, have a lower DNA methylation level (sum across all sites) than the flanking regions, as well as depleted for gene and transcription levels. See also additional example regions in Supplementary Fig. 13d, e. Source Data underlying Fig. 4a, b, d, e are provided as a Source Data file.

Given that HDF domains occupy ~60% of its genome, heterochromatin likely drives the 3D structure of chromatin folding in peppers. We observed myriad examples where prominent TAD-like domains (i.e., clearly visible as large squares in the Hi-C maps) span stretches of heterochromatin flanked by regions of active transcription (Fig. 4f and Supplementary Fig. 13d, e). These results support the hypothesis that heterochromatin and transposable elements play a central role in 3D chromatin folding in the pepper genome, and in plants more generally[12,21,22].

**TAD-like domains are often demarcated by chromatin loops.** We next attempted to annotate chromatin loops in the four studied tissues with Hi-C data combined from replicates. Using hicDetectLoops[11], we identified 5746, 5990, 7701, and 9142 chromatin loops in pulp, leaf, bud, and placenta, respectively, by merging output derived from Hi-C maps at multiple resolutions (e.g., 10, 15, 20, and 25 kb) (see Methods; Supplementary Table 13). Increased resolutions often resulted in larger loops but the vast majority (~86%) of loops identified were <2 Mb apart (Fig. 5a), which is similar to humans[8]. Approximately half of the loops identified in one tissue were detected in other tissues (Fig. 5b). Combining loops identified from all four tissues resulted in a non-redundant set containing 19,521 loops. Among them, 5728 were shared at least in two tissues and 13,793 were unique to a specific tissue. Importantly, when loops detected in one tissue were missing in another, we could not exclude the possibility that they were present but below the threshold of detection (Supplementary Fig. 15a). We reasoned that this might be due to technical limitations in loop detection approaches or reflect subtle changes in the interaction frequency between tissues[36,38]. Therefore, we also employed Mustache, which also recovers loops with high levels of confidence[64]. With Mustache, we identified 8236 non-redundant loops, of which 5282 (64.1%) were present in the hicDetectLoops calls. The set of 5282 shared loops represents a conservative set supported by both annotation methods.

In humans, chromatin loops frequently demarcate TADs—that is, the two anchors of a loop coincide with the two boundaries of a TAD[8]. Based on the set of 8,236 loops called by Mustache, we found that this pattern was also very common in the pepper genome (Fig. 5c and Supplementary Fig. 15b, c). We found that a large fraction (31.4%) of loop anchors (Mustache calls) coincided with TAD-like domain boundaries (HiCExplorer TADs identified at 10 kb resolution Hi-C map of the leaf), compared to 5% by random chance ($P$ value $<2.2 \times 10^{-16}$, Fisher's exact test). Correspondingly, ~23% of TADs had loop anchors in their boundaries, compared to 3.9% by random chance ($P$ value

$<2.2 \times 10^{-16}$, Fisher's exact test). This phenomenon can be further supported by the elevated contact frequency between the two edges of an interaction domain, which shows up as a high density in the anti-diagonal corners of domains (Fig. 5d and Supplementary Fig. 15d).

However, these loops are clearly very different from ones observed in humans[8]. First, they are outside TADs and coincide with stripes that are depleted in contacts whereas mammalian Hi-C loops are inside TADs and exhibit stripes that are enriched for contacts. Moreover, mammalian loop anchors are associated with CTCF binding sites rather than full genes. Additionally, loop anchors in pepper overlap genes twice as often as predicted by chance (~60% versus 29.4%, $P$ value $<10^{-15}$, Fisher's exact test). We frequently observe such genic loops arrayed in a sequence, with dots corresponding to mutual contacts across the entire array (Fig. 5e). Such configurations are thought to constitute transcription factories[65–67], as recently observed in the wheat genome[19]. We documented many such arrays connecting genes spaced several megabases apart in highly repetitive, gene-sparse regions (Supplementary Fig. 15e). The intergenic regions are mostly heterochromatic and tend to collapse into dense bundles of highly concentrated contacts reminiscent of TADs (Fig. 5f). Together, these observations suggest that chromatin folding bolstered by transcriptional factories may act as a common mechanism for TAD-like domain formation in pepper and perhaps most large plant genomes.

**Breaks of synteny preferentially occur near boundaries of chromatin folding domain, despite the elevated evolutionary conservation.** Given the characterization of chromatin interaction domain boundaries, we were interested in their evolutionary properties relative to non-boundary regions. We first aligned conserved syntenic sequences from potato, tomato, and eggplant to the pepper genome and found that boundaries had notably higher sequence coverage, on average than non-boundary regions, implying stronger sequence conservation (Figs. 6a and 6b, and Supplementary Fig. 16). We tested this notion further by identifying single-nucleotide variants (SNVs) and small deletions from five existing *Capsicum* assemblies -- including two cultivated *C. annuum* accessions (CM334 and Zunla-1), a wild progenitor (*C. annuum* var. *glabriusculum*), and two closely related species (*C. chinense* and *C. baccatum*) (Fig. 6a) -- relative to our CA59 assembly. Both SNVs and deletions were strongly depleted around domain boundaries (Fig. 6c); this pattern was consistent across boundaries identified from different methods (Supplementary Fig. 17a, b). Since this pattern was not observed in rice TAD-like domains[68], we

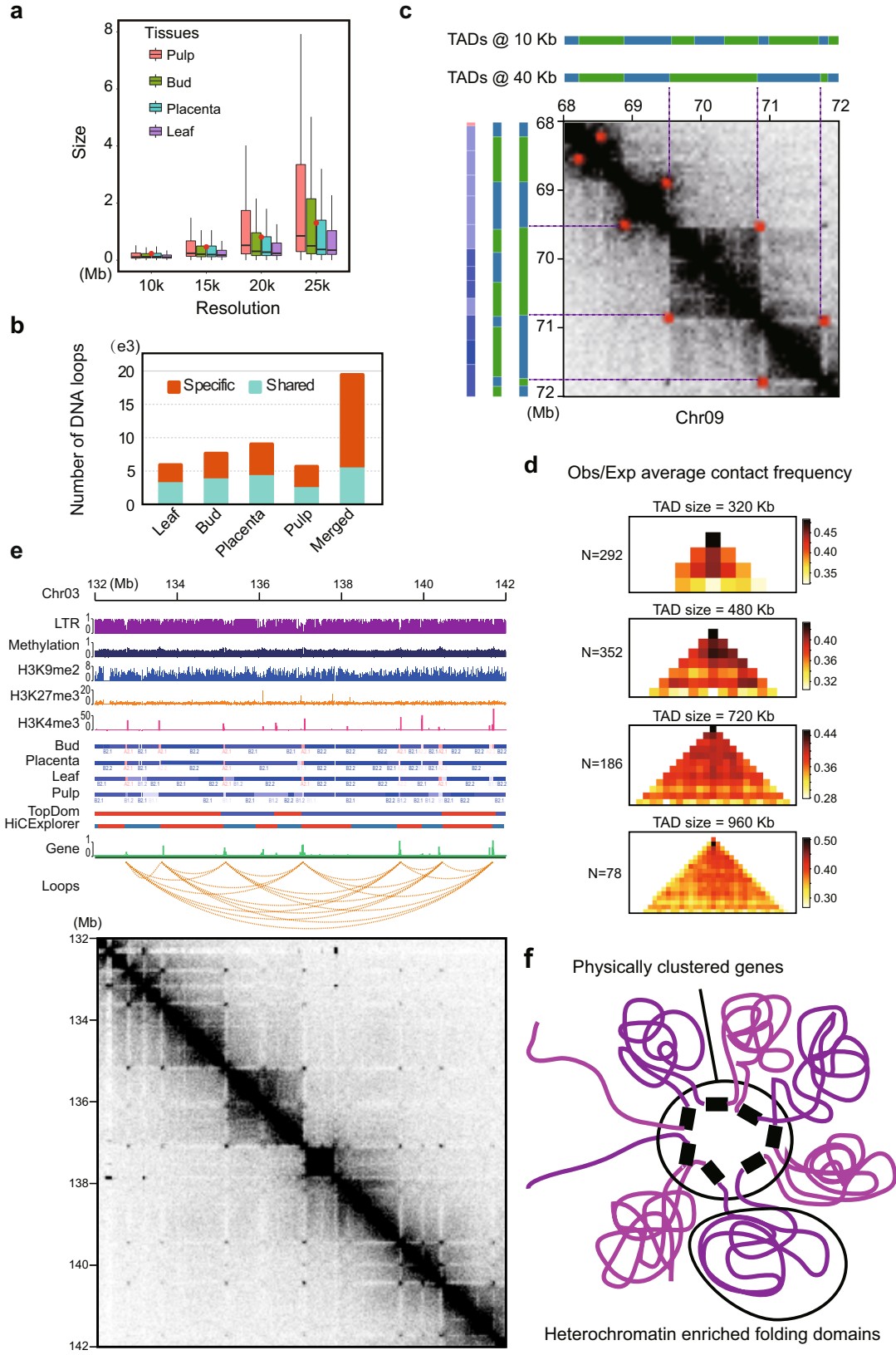

asked whether it holds outside *Capsicum*. We, therefore, analyzed *S. lycopersicum* (tomato) using published Hi-C data and genomic variant calls identified from genome assemblies of 14 *S. lycopersicum* accessions[69] (Fig. 6d). Our analysis corroborates what we observed in pepper (Fig. 6e), suggesting genomic variation is constrained around boundaries and that

this constraint may be common across Solanaceae species and perhaps beyond.

In metazoans, TADs constrain large-scale genome evolution as indicated by the observation that breaks of chromosome rearrangements preferentially occur at TAD boundaries and are depleted in TAD bodies[24,26,28]. Such a pattern, to our

**Fig. 5 Chromatin folding domains are frequently demarcated by gene-to-gene loops. a** Chromatin loops identified across tissues with Hi-C maps at multiple resolutions (e.g., 10, 15, 20, and 25 kb) by hicDetectLoops. Boxplot shows the median with (the first and third) quartiles. Red dots indicate the mean values of loop size. **b** Numbers of tissue-specific and shared loops. For each tissue, a shared loop was identified if it was present in any other tissues. A merged loop set was constructed by removing the redundant calls across all four tissues. **c** Example showing a genomic region (Chr09: 68,000,000–72,000,000) where chromatin loops demarcate TAD-like domains. Subcompartments and TADs identified at both 10-kb and 40-kb resolution for this region were shown above and right. Loops were shown as red dots in the Hi-C contact maps at 40 kb resolution (leaf). **d** Enrichment of contact frequency was observed at the corners of TAD-like domains of different sizes; that is, the peak loci are located at domain boundaries. More examples can be found in Supplementary Fig. 15d. **e** Representative example of loop anchors overlapping with genes. More examples can be found in Supplementary Fig. 15e. **f** Schematic representation of hypothesized gene-to-gene chromatin loops that mediate the formation of heterochromatin folding domains and spatial gene clusters. Source Data underlying Fig. 5a, b are provided as a Source Data file.

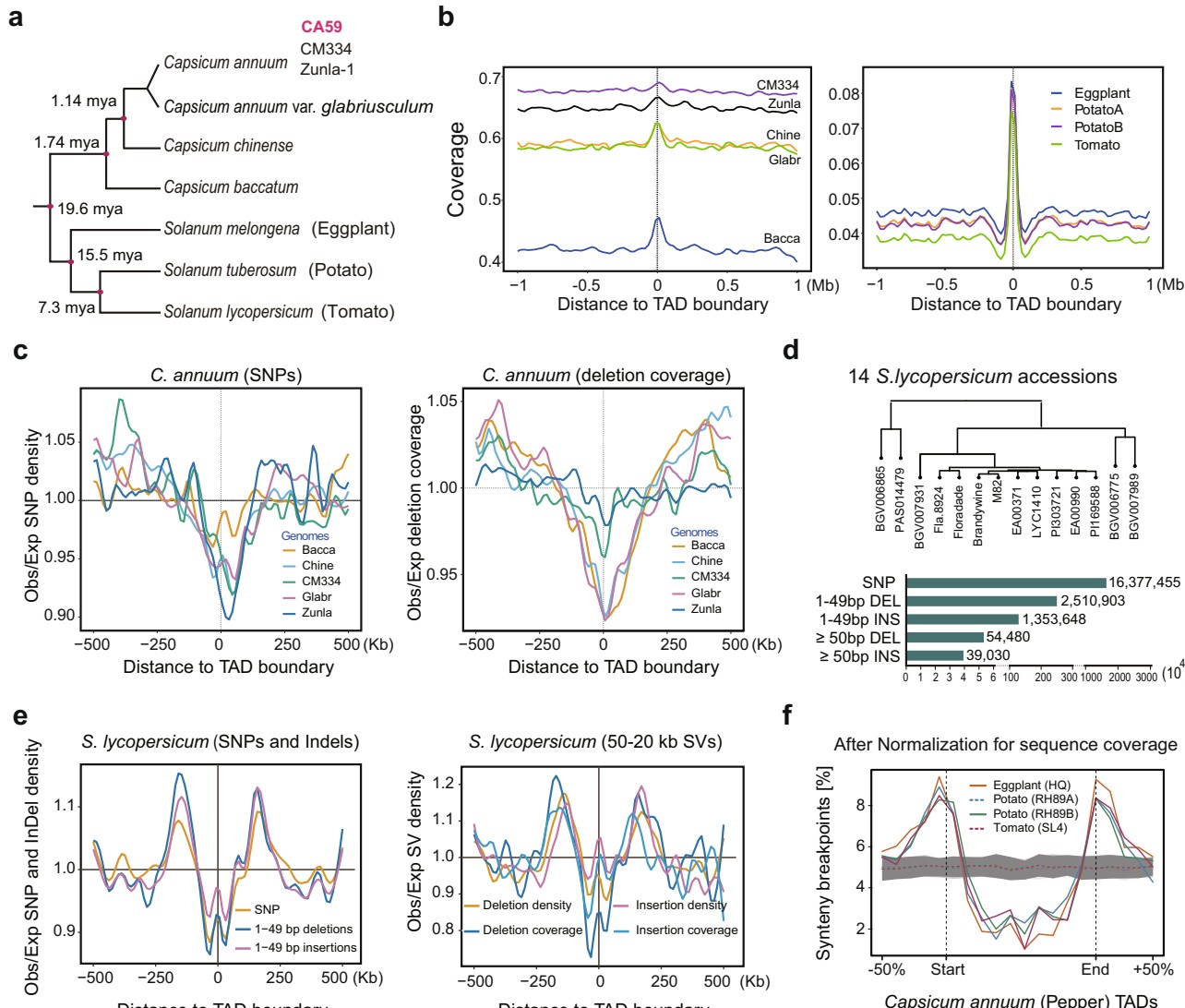

**Fig. 6 Genomic variation profiles centered at boundaries of TAD-like domain. a** Phylogenetic relationship of the studied *Capsicum* species and distantly related solanaceous species (*S. melongena*, *S. tuberosum*, and *S. lycopersicum*). The estimated divergence times were taken from previous works[49,117]. **b** Alignable fraction (coverage) of syntenic and conserved genomic sequence around pepper TAD-like domain boundaries (TopDom calls). Left is shown for comparisons between CA59 and five closely related genomes, including two *C. annuum* accessions (CM334 and Zunla-1), the wild progenitor of *C. annuum* (*C. annuum* var. *glabriusculum*), and two closely related species (*C. chinense* and *C. baccatum*). Right is shown for comparisons between CA59 and the three more distantly related solanaceous species. **c** The observed (Obs) distribution of SNPs and deletions (coverage) near domain boundaries relative to the expectation (Exp), based on the genomic background. SNPs and deletions were identified between CA59 and five closely related genomes. **d** Genomic variants identified from high-continuous genome assembly of 14 *S. lycopersicum* accessions relative to the reference genome SL4[70]. **e** The observed (Obs) distribution of SNPs, InDels, and large SVs (>50 bp) near tomato folding domain boundaries relative to the expectation (Exp), based on the genomic background. TADs were annotated by HiCExplorer with Hi-C data obtained from previous work[18] using SL4 as the reference. **f** Boundaries of chromatin folding domain in pepper are enriched for evolutionary synteny breaks identified from distantly related solanaceous species. Simulated synteny breaks data (*n* = 100) are presented as mean ± SD. Source Data underlying Fig. 6b, c, e, f are provided as a Source Data file.

knowledge, has not yet been reported in plants. To examine this question, we identified genome synteny breaks between *C. annuum* and three distantly related Solanaceae species including, *S. lycopersicum*[70], *S. tuberosum*[71], and *S. melongena*[72], which diverged from a common ancestor with *C. annuum* ~19.6 million years ago (Fig. 6a). We found that synteny breaks were indeed enriched at TAD-like domain boundaries identified for each comparison between *C. annuum* and three solanaceous species (Supplementary Fig. 17c). This pattern persisted after normalization for sequence conservation level spanning domains (Fig. 6f). We also repeated the analyses using *S. lycopersicum* and *S. tuberosum* as a reference and obtained similar results, albeit with a weaker trend (Supplementary Fig. 17d, e). These results suggest that breaks of chromosomal rearrangements are enriched at boundaries of chromatin folding domains, despite high evolutionary conservation of sequence at these regions in the Solanaceae.

**Chromatin conformation predicts transcription variance but not differential gene expression**. To explore the relationship between genome organization and gene expression, we assessed whether compartment switching is associated with transcription level in tissue comparisons. We first identified differentially expressed genes (DEGs) between pairs of tissues (see Methods) and asked whether DEGs tended to be found in regions that exhibited changed compartments (e.g., from A1.1 to A1.2). Interestingly, we failed to find enrichment in differential gene expression between tissues in regions that exhibited compartment switching (Supplementary Tables 14–17), but we did find that compartment switching modulated the amplitude of existing expression differences (Supplementary Note 3). We found that increases in subcompartment rank (e.g., from A1.2 to A1.1) were associated with increased expression magnitudes (Fig. 7a and Supplementary Fig. 18a). Conversely, decreases in subcompartment rank were associated with lower average levels of expression.

We also performed a reciprocal analysis to ask whether changes in gene expression corresponded to compartment switching. To do so, we assigned the transcribed bins (24,038 testable 40-kb bins with CPM >0.5) into three groups-that are the down and up group, in which bins exhibited expression level decreases or increases larger than twofold between tissues, respectively, and a stable group that included all other bins. We observed that although the up group contains slightly more bins with increased subcompartment rank and the down group contains slightly more bins with decreases in subcompartment rank (see Fig. 7b for comparison between bud and leaf, and the other five comparisons in Supplementary Fig. 18b), most bins in all three groups (e.g., 64.1–65.3% in the comparison between bud and leaf) did exhibit unchanged subcompartment ranks. These results suggest that changes in gene expression can only predict subcompartment switching for a small subset of genomic regions.

We next examined whether remodeling chromatin folding domains related to differential gene expression between tissues. To do so, we performed a pairwise comparison of both the chromatin folding domain profiles and the transcriptomes of the four pepper tissues. For simplicity, we divided the annotated TADs and boundaries into two groups: conserved between tissues and tissue-specific. Based on TADs annotated by TopDom, we did not detect enrichment of differentially expressed genes for either domains or boundaries in the tissue-specific group compared to the conserved group (Supplementary Table 18). However, for all pairwise comparisons between tissues, we found that conserved boundaries were associated with a lower change

level of expression than tissue-specific boundaries (three comparisons show statistically significant, Wilcoxon rank-sum test $p < 0.015$), as measured by the absolute fold change in expression level for each 40-kb bin (Fig. 7c). This pattern was not observed for domains (Supplementary Fig. 19a, b). Furthermore, when we studied the expression specificity index Tau value instead of fold change in expression level by stratifying TADs and TAD boundaries by their stability across tissues, we observed that TAD boundaries shared between/across tissues were associated with a significantly smaller variation in expression level than those unique to a specific tissue (Wilcoxon rank-sum test $p < 0.045$; Fig. 7d). As with fold change, this is not observed for TAD bodies (Supplementary Fig. 19c, d). All of these observations were confirmed with TAD annotations from Arrowhead (Supplementary Fig. 19e, f). Overall, these results suggest that TAD structures are associated with gene regulation in a way that is largely confined to genes in or near the domain boundaries.

Finally, we asked whether variation in chromatin loops is associated with changes in gene expression by comparing loops (based on hicDetectLoops inferred loops) that are shared in two or more tissues (5728) and those unique to a single tissue (13,793). Similar to results for subcompartments and TAD boundaries, differentially expressed genes were not enriched for either loop group (Supplementary Table 19). However, we found that loops shared across tissues were associated with a more stable expression level than tissue-specific loops, as shown by the fold changes in expression level (Fig. 7e) and the Tau values (Fig. 7f). These results paralleled those based on TAD boundaries. Together, our results suggest that although chromatin conformation can somewhat predict transcription variance between tissues, it does not directly determine differentially gene regulation and expression.

## Discussion

We have presented a reference-grade genome assembly for *C. annuum* and used that reference to help describe the relationship between 3D chromatin conformation, chromatin function, and gene expression. Our description has relied on extensive new Hi-C, ChIP-seq, and DNA methylation data from multiple tissues.

We first evaluated the Hi-C data, which showed that contact maps differ considerably across tissues. There are, for example, clear anti-diagonal contact patterns for leaf and bud, but these patterns are weaker or absent in pulp and placenta (Fig. 1a). These conformation contrasts may derive from tissues exhibiting differences in the so-called Rabl or non-Rabl configuration of interphase nuclei, as shown in other plant species[73,74]. These patterns are complemented by the fact that long-range (>20 Mb) interaction frequencies are enriched in leaf and bud relative to pulp and placenta (Fig. 1c, d). Despite these global differences among tissues, we nonetheless identified "A and B" compartments (Fig. 2e, f), TAD-like domains (Fig. 3c), and loops (Supplementary Fig. 15a) that were conserved across tissues. Further investigations need to illuminate the role of global chromosomal morphology and its effects on regional chromatin folding patterns[75].

By classifying the pepper genome into A & B subcompartments, we have discovered that subcompartment ranks are correlated with a series of genomic and epigenomic features (Fig. 2d, e), such as transcription levels, gene content, DNA methylation level, and intensity of histone modifications. Generally, we find that the A subcompartments have the hallmarks of active chromatin regions because they are enriched for genes, for gene expression, and for active chromatin marks like H3K4me3. Similarly, the B subcompartments appear to be more quiescent, based on higher TE content, and repressive chromatin mark

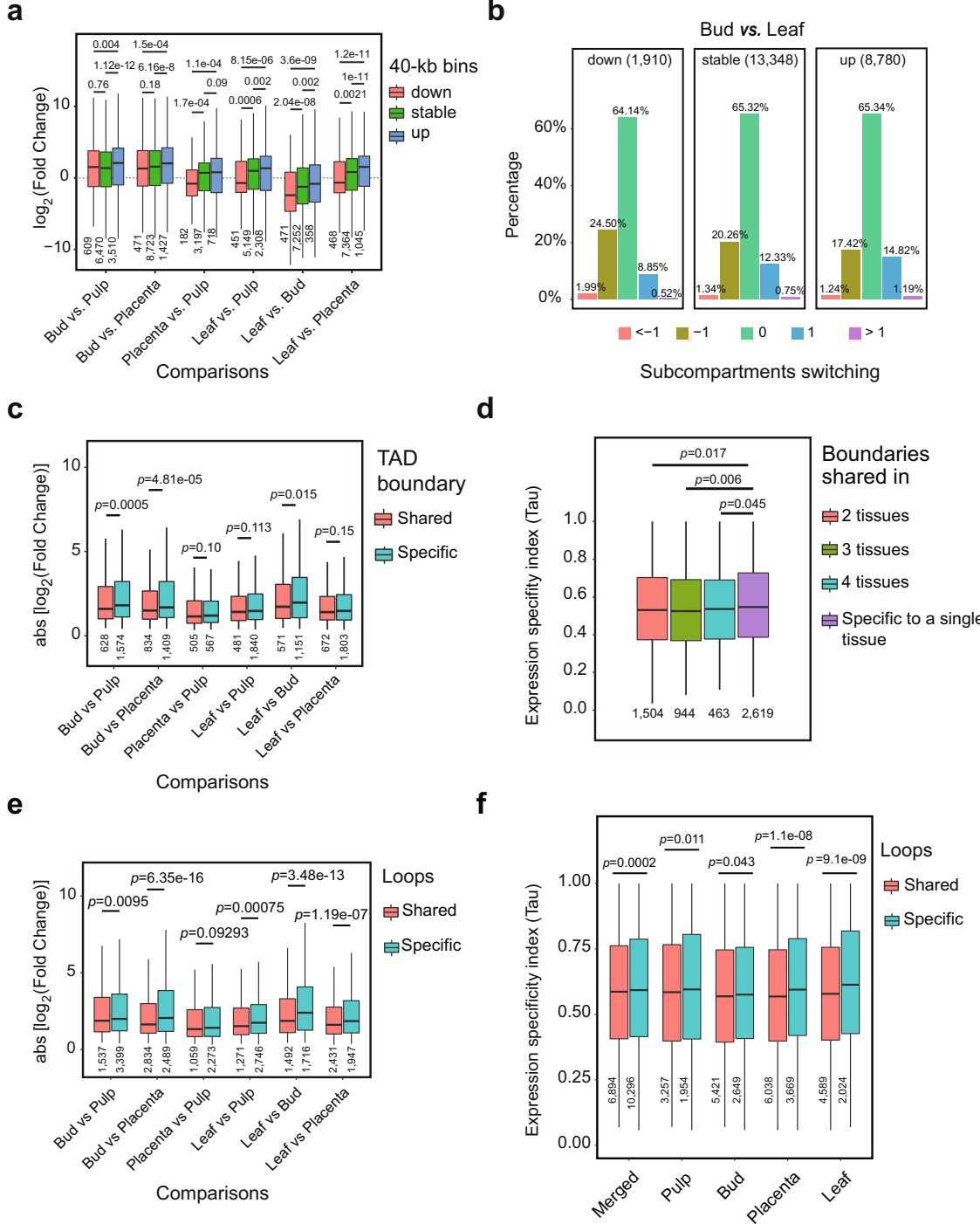

(H3K9me2). One interesting and somewhat puzzling result is the pattern of DNA methylation in and among subcompartments. The active A compartments tend to have higher methylation across the three cytosine contexts than the B compartments (Fig. 2d), but methylation in all three contexts is often associated with transcriptional repression. We suggest that this pattern reflects the prevalence of methylation features near active chromatin, including CHH islands, gene-body methylation, and more active repression of TEs. In contrast, B compartments may contain a higher proportion of fully-silenced TEs, which are often less actively targeted by methylation mechanisms. Overall, our results indicate that subcompartment identities are associated with subtle differences in genomic and epigenomic features.

We report at least three interesting features of TAD-like domains in pepper. First, they are readily identifiable, which makes pepper and other large genome plant species[12,17,18,21] superficially more similar to animals than to small genome plants like rice and arabidopsis[16,76,77]. Second, ~60% of the genome corresponds to transcriptionally repressed regions that are enriched with repetitive sequences and heterochromatin marks (e.g., H3K9me2); these regions of repressed chromatin are interspersed with active chromatin, as is seen in other plants[19] and animals[39]. Third, TAD boundaries are enriched for genes. This organization, which is similar to observations in wheat[19,20], may act to connect genes via gene-to-gene loops, leading to spatial clustering of active genes. Such configurations are consistent with the

**Fig. 7 Chromatin spatial features predict transcription variance. a** Genomic regions (i.e., 40-kb bins) switching from higher subcompartments to lower subcompartments (e.g., from A1.1 to A1.2) show a trend of decreasing expression between tissues, and conversely, switching from lower subcompartments to higher subcompartments show a trend of increasing expression. **b** Genomic regions with decreased expression were slightly enriched for cases of subcompartment switching from higher ranks to lower ranks, while those with increased expression were enriched for cases of subcompartment switching from lower ranks to higher ranks. Expression level decreases of more than twofold are labeled "down", increases of more than twofold are labeled "up", and changes within twofold are "stable". Subcompartment switching from lower ranks to higher ranks are labeled "1" if spanning 1 rank or ">1" if more than 1 rank, from higher ranks to lower ranks are labeled "−1" if spanning 1 rank or "<−1" if more than 1 rank, and unchanged are labeled "0". For more comparisons, see Supplementary Fig. 18. **c** Genomic regions overlapping with conserved TAD boundaries exhibit a relatively smaller absolute change fold in expression level between tissues than those overlapping with tissue-specific domain boundaries. Result depicted is for TopDom TAD annotation at 40 kb resolution, see results for other methods in Supplementary Fig. 19. **d** Genomic regions overlapping with shared TAD boundaries across tissues exhibit a significantly lower Tau value compared to those overlapped with tissue-specific boundaries. **e** Genomic regions overlapping anchors of shared loops between tissues have a relatively smaller change fold in expression level than those overlapping anchors of tissue-specific loops. **f** Genomic regions overlapping anchors of shared loops exhibit a significantly lower Tau value compared to those overlapping tissue-specific loops. Loops identified by hicDetectLoops were used in (**e**) and (**f**). Box plots in (**a**, **c–f**) represent the median (band inside the box), first and third quartiles. Whiskers extend to 1.5 times the IQR. Numbers below the bottom whiskers indicate the sample size. *P* values from one-side Wilcoxon signed-rank tests. Source Data underlying Fig. 7a, c–f are provided as a Source Data file.

transcription factory model[67,78] which posits that transcription factors form bridges between different genes to form transcriptional hubs.

However, it is not yet clear what genomic characteristics lead to the formation of chromatin folding domains. Previous studies have revealed that genomic features, like the physical structure of genes[28], functional noncoding sequences[79], and transposable elements or their activities[14,80], are associated with TAD structure and may facilitate their formation. The idea that sequence content affects TAD formation is consistent with the fact that boundaries tend to be near genes (Fig. 4e) and that pepper TADs are enriched in retrotransposons (Fig. 4f). Given these observations, we hypothesize that TEs play an important role in mediating the relationship between TAD-like architecture and genome size. Indeed, TADs appear as long genomic segments with higher retrotransposon density than their flanking regions. This prevalent pattern (Fig. 4f and Supplementary Fig. 13d) highlights the potential role of retrotransposons in chromatin folding. This conjecture is consistent with studies that have shown that TEs contribute to divergence and to the rearrangement of 3D chromatin organization between species[80,81]. These findings also suggest that TEs play a mechanistic role in shaping chromatin structures. If TEs do participate in organizing 3D structure, future work needs to investigate which TE features mediate this role, i.e., the relative roles of specific TE families, transcriptional activity, sequence motifs, and epigenetic effects.

One limitation of our work is that loops annotated here are based on Hi-C maps with bin sizes of 10-kb or larger (Fig. 5a), thus they may be somewhat different in the form and function from those inferred at the gene or kilobase scales using higher-resolution chromatin interaction maps[16,44,46]. Higher depth of Hi-C contact maps may be required to further decipher canonical loops in pepper, such as enhancer-promoter loops and gene loops.

Animal TADs behave as structural and functional units and can still be highly conserved between species separated by several million years[5,28,82]. In contrast, plant TAD-like domains show little conservation across distantly diverged plant species[18]. This can be partially explained by the fact that most chromatin folding domains in plant genomes are heterochromatin domains composed of rapidly changing TE content. These domains do nonetheless seem to be under some structural constraint, as evidenced by our observations that breaks of chromosomal synteny (e.g., comparisons between pepper, eggplant, potato, and tomato genomes) preferentially occur at their boundaries (Fig. 6f), similar to animals[24,26,28]. Such a pattern may be due to higher chromatin fragility at domain

boundaries and/or increased selective pressure against rearrangements that disrupt TAD-like domain integrity[25,28,83], perhaps mediated by constraints on genic co-regulation[84,85]. An interesting paradox in our observation is that, while breaks in synteny preferentially occur at TAD-like domain boundaries, these boundaries appear to be under strong sequence constraint (Fig. 6c, e), as evidenced by depletion of structural variants and SNPs in pepper, just as they are in animals[28,30,86–88]. Notably, such patterns have not been observed in rice[68], suggesting the functional and evolutionary implications of chromatin folding domains may be diverse.

The relationship between 3D genome organization and gene transcription remains an issue of open debate. Our analyses indicated that changes in chromatin spatial structures (at least for compartment, TAD-like domain, and loop investigated in this study) are not directly related to differential gene expression. This is evidenced by the observations that differentially expressed genes (DEGs) between tissues are evenly distributed across the genome irrespective of where chromatin features changed or not (Supplementary Table 14–19). In addition, genomic regions with DEGs are also always associated with unchanged 3D genome organization (Supplementary Fig. 18b). However, we have observed that the preservation of spatial chromatin features is subtly associated with gene expression stability across tissues (Fig. 7a–f), which has also been illustrated previously[24,28,36,38]. Taken together, these findings continue to suggest that genome architecture broadly (but subtly) affects patterns of gene expression. Recent works[40–43] have bolstered this claim by suggesting that chromatin conformation provides a structural scaffold for the establishment of the regulatory environment in the nucleus.

In summary, we integrated genomic, transcriptomic, and epigenomic data to create a 3D chromatin map of the pepper genome. We also provided a preliminary mechanistic explanation of chromosome folding in this large (~3 Gb) plant genome. Our results suggest that heterochromatin-driven folding is a foundational force shaping pepper genome organization, resulting in TAD-like domains that cover ~60% of the genome. In addition, loops formed between genes via transcription factories may facilitate such folding architecture. We also showed that the spatial genome structures of pepper and its relatives are under structural and sequence constraints similar to those documented in animals. Nevertheless, there remains much to be explored regarding the structural and mechanistic bases for chromatin structures. Such an understanding will serve as a guide for sequence-based modeling and targeted engineering of the 3D genome[89,90].

## Methods

**Plant materials and DNA sequencing**. The pepper (*C. annuum*) inbred line, designated as "CA59", was used in this study due to its desirable agronomic characteristics, including high yield, broad-spectrum disease resistance, and abiotic stress tolerance[47]. Seeds were germinated in the soil in 72 cell plastic flats and placed in the greenhouse on February 2nd and August 6th. The seedlings were grown in a greenhouse under normal conditions in Guangzhou, China (23.1291° N, 113.2644° E).

Thirty-day-old fresh leaves harvested from a single individual plant were used for DNA extraction and sequencing. For BGI (Beijing Genomics Institute) short-read sequencing, DNA was extracted from about 2 g leaves using a modified cetyltrimethylammonium bromide (CTAB) method[91]. A sequencing library with an insert size of 350 bp was prepared using the VAHTS Universal DNA Library Prep Kit (Vazyme, Nanjing, China). Quality assessment of the library assessing DNA quantity, purity, and size range was conducted using Agilent Bioanalyzer 2100 (Agilent Technologies, Santa Clara, CA). The library was sequenced on the MGI-SEQ 2000 sequencing platform to produce pair-end sequence data (2 × 150 bp). For Pacific Biosciences (PacBio) sequencing, extraction of high-molecular-weight DNA was carried out as above[91]. About 10 μg of genomic DNA was used to prepare template libraries of 30–40 kb using the BluePippin Size Selection system (Sage Science, USA) following the manufacturer's protocol (Pacific Biosciences, USA). The libraries were sequenced on the PacBio SEQUEL II platform with three SMRT flow cells.

**Genome assembly and quality assessment**. Genome size was estimated using G.C.E. (Genome Characteristics Estimation) (1.0.2)[92] with parameters: -m 1 -D 8 -b 0 -H 1 (Supplementary Method 1). G.C.E. calculated the 17-mer frequency distribution based on 362.0 Gb cleaned BGI short reads. The estimated genome size is about 2.95 Gb and the heterozygous rate is 0.23%.

We obtained 451.9 Gb clean PacBio long reads (~153× genomic coverage) from three SMRT flow cells with a subread N50 of 28,351 bp. To perform a de novo assembly, we first filtered out short-length PacBio raw reads and only retained the top 200 Gb longest reads (which with a subread N50 of 39,818 bp and ~66× genomic coverage) for correction using MECAT2 (v20200228)[93]. The corrected reads were then trimmed and assembled using CANU (2.0)[94]. The initial contig assembly was further polished through three iterations using Pilon (1.23)[95] with ~123× BGI short reads. Finally, we used the Juicer(1.56), Juicerbox(1.11.08), and 3D-DNA pipeline(180114)[52,96,97] with 415.2 Gb Hi-C data (~141× genomic coverage) from bud and leaf to build scaffolds, following manual correction. For more details, see Supplementary Method 2.

The Phred quality score QV was computed as $-10\log_{10}(P)$, where P indicates the probability of error. This error rate (P) was calculated by dividing the sum of all variant sites (SNPs and InDels) from mapping BGI reads to the assembly to the total size (only for sites covered by at least 3 reads) of the assembly[98]. BUSCO (3.0.2)[99] score was used to evaluate the gene-space completeness based on Embryophyta odb9 dataset (*n* = 1440). Synteny dot plots between CA59 assembly and other related genomes (e.g., *C. annuum* cv Zunla-1[48], tomato, potato, and eggplant) were performed using Minimap2 (2.17)[100] and PAFR (https://github.com/dwinter/pafr, version 0.0.2).

**Transcriptome sequencing**. Long-read full-length transcriptome sequencing was performed for five tissues, including leaf, bud, placenta, root, and pulp, using the PacBio isoform sequencing (Iso-seq) platform. Between 35,257 and 50,237 full-length transcripts (Supplemental Table 5) were assembled across tissues using the SMRTlink (version 8) pipeline (Supplementary Method 3). These transcripts were used for guiding gene annotation. Additionally, RNA-seq data for the corresponding tissues, each with three biological replicates, were collected (Supplementary Tables 7, 20). Pooled tissues from five individual plants were used for RNA extraction. See more details for RNA-seq protocol and data processing in Supplementary Methods 4, 5.

**Transposable elements and gene annotation**. TEs were annotated with Extensive de novo TE Annotator (EDTA) (1.9.6)[101]. Gene models were annotated using MAKER (3.01.03)[102], which was performed in three iterations. To run MAKER, TE library derived from EDTA, Iso-seq full-length transcripts from five tissues, and gene models from the Zunla-1 assembly[48] were used as supportive evidence to guide the prediction of gene models. Additionally, RNA-seq-based transcripts were constructed for each tissue using the HISAT2 (2.2.1)[103] and the StringTie (2.1.4)[104] pipeline. These predicted new transcripts were merged with the MAKER gene models to produce the final gene/transcript set. For more details, see Supplementary Methods 6, 7.

**Hi-C experiment, sequencing, and data processing**. We generated in situ Hi-C data for four tissues, including leaf, placenta, pulp, and bud, each with two biological replicates (Supplementary Tables 7, 20). Hi-C libraries were constructed according to the protocol established by Rao et. al.[8] (Supplementary Method 8). Sequencing was performed (150 bp paired-end) on the MGI-seq 2000 platform. Public Hi-C data used in this study are listed in Supplementary Table 21. Hi-C raw reads were cleaned using Trimmomatic (0.38)[105]. Hi-C contact maps were

constructed using both Juicer (1.5.6)[52] and HiCExplorer (3.53)[11] pipelines (Supplementary Method 9). Quality and reproducibility of the Hi-C data were assessed using the QuASAR-Rep scores calculated by 3D Chromatin-ReplicateQC (0.0.1)[53] and a Pearson correlation analysis with the HiCExplorer tool hicCorrelate.

**Bisulfite sequencing and data processing**. DNA was isolated from leaf tissue harvesting from 30 days old plants of the CA59 accession to generate bisulfite sequencing (BSseq) data. Bisulfite libraries for two replicates were prepared and sequenced on (150 bp paired-end) the Illumina Novaseq 6000 system (for more details, see Supplementary Method 10). To measure the genome-wide DNA methylation level, BSseq reads were first trimmed for quality and adapter sequences using Trimmomatic (0.38), resulting in a total of 90.34 Gb (29.4x genomic coverage) and 85.23 Gb (27.8x) clean reads in two replications, respectively. Bismark[106] (0.23.1), in conjunction with bowtie2 (2.4.4) was then used to align the trimmed reads to the genome. The number of methylated and unmethylated reads per cytosine was determined using the Bismark bismark_methylation_extractor tool. DNA methylation level of a genomic bin (e.g., 10 and 40 kb) was computed as the percentage of heavily methylated sites (which we defined as methylated reads contributing to at least 25% of all mapped reads) in that bin. This value was calculated for three contexts (i.e., CpG, CHG, and CHH), separately. Also, a value that we defined as the overall methylation level was calculated by summing across all three contexts.

**ChIP-seq and data processing**. Chromatin Immunoprecipitation sequencing (ChIP-seq) data of three histone modifications (including H3k4me3, H3K27me3, and H3K9me2) was generated for leaf tissue of the inbred line CA59 (Supplementary Table 7). Two replicates were performed for each mark. ChIP-seq experiments and library preparation protocols were described in Supplementary Method 11. Sequencing (150 bp paired-end) was performed on the Illumina Novaseq 6000 system. To measure the genome-wide profiles of the investigated histone modifications, ChIP-seq raw reads were trimmed for quality and adapter sequences using Trimmomatic (0.38) and were then mapped to the CA59 genome using BWA (0.7.17). Enrichment peaks were called using MACS2 (2.2.7.1) to verify the quality of ChIP-seq data. The ChIP-seq intensity of a genomic bin (i.e., 10 and 40 kb) was calculated using the bamCoverage tool from DeepTools (3.3.0) with the read coverage normalized in CPM.

**Compartment identification and analysis**. To identify the A and B compartments, we first adapted the PCA-based method[1,37]. Briefly, the observed/expected matrices were first calculated with normalized and corrected (ICE) interaction matrices for each chromosome at 500-kb resolution. Next, Pearson correlation and covariance matrices were computed on the observed/expected matrices. Third, PCA eigenvectors were calculated with the covariance matrices and the first principal component (PC1) was used to assign the A and B compartments according to the direction of the eigenvalues which were manually adjusted by the gene and TE density. All these steps were processed using HiCExplorer[11]. This method is capable of identifying the A and B compartments globally when working on Hi-C maps at the 500-kb resolution, whereas it failed to identify the A and B compartments consistently when using Hi-C maps at a relatively higher-resolution (e.g., 40-kb).

To further characterize regional compartments at a finer resolution, we used Calder[13] to infer subcompartments based on Hi-C matrices at 40-kb resolutions. To do this, the HiCExplorer interaction matrices were first transformed to a square format and were then imported into the R package, BNBC[107], for normalization and batch correction across tissues and replicates. Next, the corrected matrices were converted into a three-column format which is required as input for Calder. We ran Calder with the default parameters which infer 2, 4, and 8 subcompartments, at three hierarchical levels. Each hierarchical level contains an equal number of "A" and "B" subcompartments. For example, in the eight subcompartments level, 4 (designated as A.1.1, A.1.2, A.2.1, and A.2.2) belonging to the A compartment and the other four (designated as B1.1, B.1.2, B.2.1, and B.2.2) belonging to the B compartment. We also repeated the above analysis for the 10-kb resolution matrices. Because 10-kb resolution matrices were generated by combining Hi-C data from replicates, we omitted the BNBC correction process.

**TAD annotation and classification**. We tested and compared three tools, including HiCFindTADs[11], TopDom(0.0.2)[108], and Arrowhead[8], to identify TADs. To assess the reproducibility between TADs identified by different tools, we first applied them to a leaf Hi-C interaction matrix at 40 kb resolution. We measured the similarity by comparing TAD bodies, TAD boundaries, and genome coverage of conserved TADs derived from different tools. For TAD bodies, a reciprocal overlap threshold of >80% of genomic coordinate was used to define conserved TADs between tools. For TAD boundaries, a conserved call between tools was considered if the genomic intervals of the boundary overlap with each other or are apart less than one bin size, i.e., 40 kb. Additionally, we assessed the performance of TADtool[59], which is based on the insulation index algorithm, on TAD calling and compared it with the above tools (Supplementary Note 2).

To assess the similarity of TADs between samples, we chose to use TopDom, because of its top performance in a previous benchmarking study[60] and the inferred

TADs are compatible with the continuous distribution manner of TADs in the pepper genome. Furthermore, its input files are compatible with other processing programs, for example, BNBC[107] which was used to normalize and correct Hi-C matrices. Hierarchical clustering analysis was used to explore the similarity between samples based on the Jaccard distance ($J(A, B) = |A \cap B|/|A \cup B|$, where A indicates TADs annotated in one tissue and B in another tissue) calculated based on the genome coverage of shared TADs between samples. Alternatively, the percentage of shared TADs, their boundaries, and their genome coverage between samples were used as similarity distance in the hierarchical clustering analysis for comparison. The TAD set from TADtool was compared as well.

To classify TADs, we calculated the Euclidean distances between TADs (TopDom set) based on the similarity of their genomic (TE and gene content) and epigenomic features (overall methylation level, intensity of H3K4me3, H3K9me2, and H3k27me3 marks). Then, the hierarchical clustering analysis of TADs based on the calculated Euclidean distances was performed using the hclust function with the 'complete' method in R (4.0.4) with heatmaps constructed using the heatmap.2 function. Hi-C contact maps were displayed using hicPlotMatrix from the HiCExplorer tool and the online JuiceBox tool https://aidenlab.org/juicebox/.

**Identification and analysis of genomic variants.** Genomic variants (e.g., SNPs, 1–49 bp InDels, and >50 bp structural variations) between genome assemblies were identified using a custom pipeline[28] which includes four key steps: (1) genome-wide local alignment with Minimap2 (2.17)[100]; (2) building alignment chains using the chain/net/syntenic workflow[109]; (3) identifying genomic variants between a pair of genomes; and (4) genotyping genomic variants in multiple genomes. Using this pipeline, we inferred genomic variations for 14 tomato genomes relative to the reference SL4 (Fig. 6d). We also identified SNPs and genomic coverage of deletions from five closely related genomes relative to the CA59 genome, including two within-species accessions, CM334 and Zunla-1, a wild progenitor, *glabriusculum*, as well as two closely related species, *C. chinense* and *C. baccatum*, using a custom Perl script PairwiseGV.pl.

To measure the relative abundance of genomic variants around boundaries of chromatin domains (TADs), we used a sliding window approach with a bin size of 40 kb and a step size of 5-kb to generate an observed/expected matrix within 500 kb of the boundaries. We assumed that the genomic variants are homogeneously distributed along the pepper genome. Insertions were excluded in the analysis of pepper genomes due to assembly quality issues.

**Identification and analysis of synteny breaks.** We used a custom Perl script PairwiseSynteny.pl to identify synteny breaks between a pair of large plant genomes by parsing the *.syntenic* file obtained from the above minimap2/chain/net/ syntenic workflow. Synteny breaks were identified from all pairwise comparisons among four distantly related Solanaceae genomes (e.g., pepper, tomato, eggplant, and potato) with each as a reference except eggplant.

We quantified the distribution of evolutionary synteny breaks along with the chromatin domains (TADs)[24]. We noted that TAD boundaries are enriched for evolutionary sequence conservation which might result in an enrichment of synteny breaks identified in such regions. We reduced the impacts of this bias by normalizing the observed distribution with the rate of alignable sequence between genomes along with the TAD bodies. Significance tests were performed by simulating 100 random sets of synteny breaks for each comparison.

**Loop identification and analysis.** Chromatin loops were annotated using the hicDetectLoops tool from HiCExplorer (3.5.3)[11]. To obtain denser Hi-C interaction matrices, we combined Hi-C data from two replicates for each tissue. Hi-C interaction matrices were normalized using the Knight-Ruiz (KR) method. Because the visual inspection of Hi-C contact maps shows extensive loops that can span over several megabases, we called loops from Hi-C interaction matrices at multiple resolutions, including 10, 15, 20, and 25 kb. Loops identified from all resolutions were then merged within 25 kb to produce the final loop set using hicMergeLoops from HiCExplorer. We also used Mustache[64] to call loops with the Juicer Hi-C interaction matrices. We used the Intersect function from pgltools (2.2.0)[110] with the parameter: "-d 25 kb" to determine if loops are shared between tissues. Parameters used for loop calling were listed in Supplementary Table 13.

**Overlap between TADs with compartments and loops.** To estimate the extent to which chromatin domains (TADs) overlap Calder-inferred subcompartments, we performed multiple pairwise comparisons between TADs and subcompartments that were both inferred with different conditions, respectively. For example, TADs were annotated using HiCExplorer, TopDom, and Arrowhead using both 40-kb and 100-kb resolution matrices, with an additional set called from 10-kb resolution matrices using HiCExplorer. Subcompartments were annotated at 10-kb, 40-kb, and 100-kb resolution matrices. A reciprocal overlap threshold of >80% of the genomic coordinate was used to determine whether they coincide with each other.

To assess the frequency of TADs that are demarcated by loops--that is, two boundaries of a TAD coincide with the two anchors of a loop, we constructed a PGL file by pairing TAD boundaries sequentially. The Intersect tool in the pgltools (2.2.0) was then used to determine whether TADs overlap with loops.

**RNA-seq analysis and expression patterns.** RNA-seq data quality control and processing were conducted as described above. Gene expression was quantified in normalized TPM (Transcript Per Million) using FeatureCounts (2.0.1)[111]. Expressed genes were defined as those with CMP >0.05. Of 38,974 expressed genes, between 6974 and 17,576 across pairwise comparisons between tissues were identified as differentially expressed genes (DEGs) using the Limma (3.46.0) package[112] in R with an adjusted $P$ value < 0.01 (Supplementary Table 14).

To facilitate the correlation analysis of transcription and chromatin features (which are generally annotated with a fixed genomic size, e.g., 40 kb), we calculated the number of reads per 40-kb bin (coverage tracks) from RNA-seq alignments using the bamCoverage tool from deepTools (3.3.0)[113] with the following parameters: "--binSize 40000 --minMappingQuality 30 --outFileFormat bedgraph". RNA-seq data from five tissues were used to broadly characterize the expression pattern for 40 kb bins. We measured two properties of the expression pattern: expression fold change and tissue specificity index tau[114]. The expression level for each bin was normalized in CPM (counts per million). The expression fold changes and differentially expressed bins between tissues were obtained using the Limma-Voom package[112,115] in R. Tissue specificity was calculated using the formula $tau = sum (1-r_i)/(n-1)$, where $r_i$ represents the ratio between the expression level in sample i and the maximum expression level across all tissues, and $n$ represents the total number of tissues. The value of tau ranges from 0 to 1, with higher values indicating greater variation in expression level across tissues, suggesting higher tissue specificity. Expression values were averaged among three replicates.

**Correlation analysis between chromatin spatial organization and transcription.** To evaluate whether subcompartment switching is correlated with changes in expression level, we classified all 40-kb genomic bins into three groups based on changes in subcompartment between tissues: (1) the "down" bins in which sub-compartments transitioned for at least 1 scale in the order from A1.1 to B2.2; (2) the "up" bins in which subcompartments transitioned for at least 1 scale in the reverse order; and (3) the "stable" bins in which subcompartments remain unchanged. We first tested whether the "down" and "up" bins overlap with more DEGs and then correlated this classification with fold changes in expression level between tissues which were calculated above. We adopted the following approaches for dealing with replicates when it is necessary: (1) we evaluated the consistency of results between independent analyses done for each replicate; or (2) used only the chromatin features shared by both replicates for analyses.

To evaluate whether a reorganization of chromatin domains (TADs) and loops is correlated with changes in expression level, we classified TADs, TAD boundaries, and loops each into two categories: (1) tissue-specific group, in which they were identified in only one tissue, and (2) conserved group, in which they were identified in two or more tissues. We correlated this classification of TAD features and loops with the changes of gene expression profiles, including DEGs and fold changes between tissues, as well as the tau values calculated above.

**Reporting summary.** Further information on research design is available in the Nature Research Reporting Summary linked to this article.

## Data availability

The data that support the findings of this study are available within the paper and its Supplementary Information files. A reporting summary for this article is available as a Supplementary Information file. The raw sequence data and genome assembly (CA59) have been deposited into CNGB Sequence Archive (CNSA) of China National GeneBank DataBase (CNGBdb) with accession number CNP0001129 and National Center for Biotechnology Information (NCBI) with project accession PRJNA788020. All accessions of published Hi-C data used in this study are listed in Supplementary Table 21. Source data are provided with this paper.

## Code availability

All scripts that reproduce analyses from the manuscript are available on GitHub [https://github.com/yiliao1022/Pepper3Dgenome][116].

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

## Acknowledgements

This work was funded by the National Natural Science Foundation of China (32070331, U21A20230, 32102380, and 32072580), the Guangdong Basic and Applied Basic Research Foundation (2020A1515011396), the National Key Research and Development Program (2018YFD1000800), and Strategic Priority Research Program of Chinese Academy of Sciences (XDPB16). This work was jointly funded by National Institutes of Health (R01GM123303), National Science Foundation (IOS-1656260), and start-up funding from the University of California, Irvine to J.J.E. We are grateful to Dr. Shujun Ou (Johns Hopkins University, USA) and Hongru Wang (University of California Berkeley, USA) for discussions and advice on the project. We also thank Frasergen Bioinformatics Co., Ltd (Wuhan, China) and Shanghai Jiayin Biotechnology Co., Ltd for Tech & Expert Services.

## Author contributions

Yi.L., J.J.E., and C.C. jointly supervised this work; Yi.L. conceived of the presented ideas; C.C., Z.Z., J.L., F.L., and B.C. carried out the experiments and obtained the resources; J.W. assembled and annotated the genome; Yi.L. analyzed the data and wrote the original draft with input from J.W. and C.C.; J.J.E., B.S.G., J.C., Y.Z., and Yuanlong L. reviewed and edited the draft; all authors discussed the results and commented on the manuscript.

## Competing interests

The authors declare no competing interests.

## Additional information

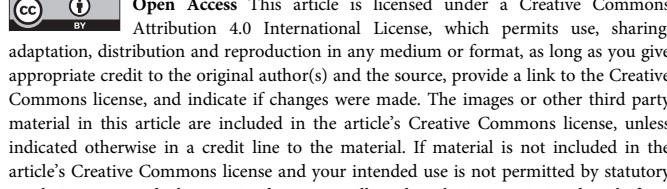

