## [Peer Review File · Nature Communications]

The 3D architecture of the pepper genome and its relationship to function and evolutionReviewers' Comments:

Reviewer #1:

Remarks to the Author:

In this paper, Liao et al have performed extensive HiC on pepper and have written a report on their findings. Their main findings is that TADs are prominent in pepper, oppositely to other plant species. They also find (sub)compartments that correlate with transcription and compare the 3D organisation between different tissues finding some differences.

Interestingly, their most prominent TADs are the ones that are enriched in transposons.

They also claim that loops are present at the TAD boundaries and that they may be caused by loop extrusion.

Finally, they relate their findings to synteny and conserved sequences finding that breaks of synteny appear at TAD boundaries.

Overall, I think this is a careful and well-executed work that has required a lot of effort. I appreciate the fact that the authors give a lot of details on their experimental and computational analysis and have done the bioinformatics using different complementary tools.

At the same time, I also feel that some of their results may be due to misinterpretation of their data. I think before publication in Nat Comm, the authors should significantly revise the paper and strengthen (or drop) some of their claims.

Comments to address:

1. I typically would not recommend to add more work to a paper that contains already a lot, but this paper clearly lacks some chip-seq. The inactive TADs made up by transposons sound to me like heterochromatic regions that are dense and collapsed. (The absence of gene expression is somewhat already confirming this hypothesis). This picture would thus be very different from the typical TAD in, e.g. humans, where there may not be a single clear epigenetic mark (as that would be considered a compartment).

1a. Given the point above, it is not clear to me why TADs and compartments do not overlap as most of the inactive transposon-rich TADs should perfectly coincide with B compartments. Can the authors comment on this?

1b. It looks to me that most of the discussion evolves around the interpretation of HiC maps which is done using different tools, over different tissues or related species. Interpreting HiC maps is notoriously difficult and I don't blame the authors for elaborating at length on this but I have the impression that they sometimes misinterpret the maps (see point 2) and/or they "see too much into them".

1c. Related to the point above, I don't understand what's the point in classifying the compartments in 8 subcompartments. Why not 6 or 4? Why not leaving 2 (A/B) compartments? What do we learn from this further subclassification? My interpretation of Figure 2f is that this classification is an overfitting of the PCA.

1d. The authors mention that "However, this method failed to resolve the identification of A/B compartments using Hi-C contact matrices at higher resolution (e.g. 40-kb)." Why is this A/B classification failing? Maybe there is some interesting biology/biophysics hidden behind this failure. After all, standard A/B PCA works well in high resolution data (e.g. human HiC at 1kb resolution).

1e. I find rather strange that 50% of the genome is in the A (transcriptionally active) compartment (line 124). Given the large amount of transposons I would have expected most of the genome to be in the B (silenced) compartment.

2. TAD loops - the authors claim that they observe loops in their HiC maps. This is true, but these loops are clearly very different from the ones observed in humans by Rao et al or other works. First, these loops are *outside* the TAD and coincide with horizontal and vertical stripes that are "depleted" in contacts. On the contrary, the dots on mammalian HiC are *inside* the TADs and coincide with stripes that are "enriched" in contacts. Additionally, the mammalian dots do not coincide with full genes and instead match with CTCF binding sites. (This connects to my previous comment, i.e. lack of chipseq in this work.)

As shown in Fig.5e, it is clear that in pepper, these spots in the HiC map are clusters of genes. Since the genes are short, they appear as "spots" or "loops".

2a. A simple model to explain these "loops" of genes is the transcription factory model by Peter Cook and coauthors or the transcription hubs/clusters (or even condensates) models (See I Cisse, R Young, etc). In these models transcription factors form bridges between different genes and glue them together in hubs.

No need of loop extrusion to explain these HiC features. See

<https://academic.oup.com/nar/article/44/8/3503/2467087>

<https://www.nature.com/articles/s41467-021-25875-y>

2b. From Fig.5e is also clear that the PC analysis and TAD calling struggle to capture these small islands of active genes at this resolution. They are identified as TAD boundaries but are effectively small compartments upon closer visual examination.

2c. I strongly suggest the authors to revise their text around the statements of TAD loops and loop extrusion and/or provide stronger evidence for a mechanism (loop extrusion or others) for these features. For instance "predominantly partitioned into TADs and that a large proportion of TADs are likely formed by loop extrusion..." needs to be removed as it is likely wrong. SMC chipseq data is otherwise needed to support this statement.

3. I am rather impressed by the similarity of small-scale 3D genome organisation (TADs and features near the diagonal) across tissues (Fig.3c). The leaf clearly displays a Rab1-like long-range contacts but apart from this, the small scale arrangement doesn't seem too different for the other tissues. This suggests that TADs may not be needed to control gene expression, as mentioned by the authors. Can the authors probe this further? Are there genes that can be turned on in response to external stimuli such as heat/cold? Can they measure (via HiC, or 4C) the change in organisation upon activation?

4. Again on TADs/compartments: I disagree on the statement at line 591 "One striking ... more defined TADs". I would like the authors to think about the definition of TAD and that of compartments. At what point a compartment is a TAD and viceversa? Figure 3a is an example of this, the checkerboard pattern on the left half of the panel is a signature of a compartment that can be made by bridge proteins binding specific epigenetic marks (See again <https://academic.oup.com/nar/article/44/8/3503/2467087>). Yet, the squares near the diagonal will be wrongly classified as TADs.

I suggest the authors to critically think about their results - why should a larger genome size imply the existence of TADs? Is it not more likely that Pepper displays (in accordance of other plants) a lack of TADs and that the ones seen in this work are instead wrongly classified?

Indeed, by staring at extended data Fig.1, it is not clear to me that pepper HiC map is different from the rice one, apart from that the rice displays smaller domains (but it is also true that the scale is different in the map ~3Mbp VS ~20Mbp).

4a. The potential pitfall in highlighting the presence of TADs in this work is also clearly seen in the fact that 3 different tools yield substantially different TAD annotations (extended data fig 7). Which one should we trust?

4b. On a global level, it may seem that pepper has more longer range contacts than the other plants. Can the authors plot the contact probability of pepper against that of the other plants? What's the difference in volume fraction occupied by the genome in the nuclei of these species? Is pepper's genome much more crowded/confined than the genome of other plants?

5. I don't understand how can TADs not coincide with a subcompartment (line 616, "only 11% of TADs coincide with ..."). From what I understand, each genome segment is assigned to a (sub)compartment, so by definition each TAD should contain one or multiple (sub)compartments. Can the authors clarify?

6. Line 618. Another example of misleading interpretation of the evidence: "Based on these ... HiC maps". First, these HiC maps show no evidence of loop extrusion (the spots are not inside TADs but are made by neighboring active genes that are clustered). Second, a simpler explanation is the heterochromatic collapse of silenced regions.

Reviewer #2:

In this manuscript, the authors assembled a *Capsicum annum* genome and constructed Hi-C maps for four tissues, including leaf, bud, pulp and placenta. The title of the manuscript is “The 3D architecture of the pepper (*Capsicum annum*), genome and its relationship to function and evolution”. However, I did not find any specific analyses related to function, and the analysis of evolution is only reflected in a section “Breaks of synteny preferentially occur near TAD boundaries, despite high evolutionary”. Overall, many parts of this manuscript gave only some descriptive results from data analysis, without functional study of at least one chromatin structure with specific functional implications. The authors obtained some general conclusions about 3D genome organization in pepper, but most of them were previously known in other species. My other comments are shown below:

1. Please explain ‘Phred Quality Score of Q52’ in line 128 and ‘EBP 6.C.Q40 standard’ in Line 144
2. The intact LTR-RTs have two peaks (~0.2 Ma and ~2 Ma) in the pepper genome. Are there additional peaks if truncated LTRs are included for analysis? Because 2% of repeats cannot represent the LTR burst time of the entire genome.
3. Previously published genome and the current assembly should be compared. This may provide some evidence showing the high quality and improvement in the current assembly.
4. References in Line 97-98 showing “recent studies in wheat have reported that chromatin loops demarcate a large proportion of chromatin domains” should be added.
5. Did the authors performed a sequencing depth analysis for Hi-C data in different tissues? Different depths regarding of the number of valid interactions have an impact on the comparison between different tissues.
6. Why not show corrected Hi-C counts from 0-~10⁶ in **Fig. 1c**. The unit of the x-axis should be added.
7. Why did the authors use CPM instead of FPKM or TPM to represent the expression levels of genes? The CPM value is correlated with the gene length.
8. Abbreviations of the units that indicate chromatin length in figures include Mpb (Fig. 1b, Fig. 2f, Fig. 3c, Fig. 3d, Fig. 4a, Fig. 4f), Mb (Fig. 3b, Fig. 4c, Fig. 5a, Fig. 5e), kbp (Fig. 5d), kb (Fig. 4h, Fig. 2d), Kb (Fig. 6e) and none (Fig. 1c, Fig. 3a). It is better to be consistent.
9. Some TADs are not conserved/consistent in four tissues and some are not prominent in **Fig. 3c**.
10. “In all comparisons, we found less than 11% of TADs coincide with CDs. This result suggests that compartmentalization may contribute to the formation of only a small fraction of TADs in the pepper genome” in **LINE 296-299**. I believe that the overlap

between two structures is related to the size of the structure. If the authors characterize the overlap ratio between TADs and compartment, this value may be higher than 11%. I suggest that more details should be shown to support the conclusion that compartmentalization may contribute to the formation of only a small fraction of TADs in the pepper genome.

11. Using a 40-kb resolution leaf Hi-C map, these three methods identified 1,780, 4,663, and 2,641 TADs in **LINE 264-265**, but the sum of TADs in **Fig. 4b** is 2640.
12. The authors mention that “the most prominent TADs formed in genomic intervals with a long stretch of enriched LTR-RTs and were always flanked by active transcription regions (Fig. 4f and Extended Data Fig. 8d)” in **LINE 319-321**. It is not enough to use two examples to illustrate this phenomenon and more statistics are needed to support it.
13. **Fig. 6c** shows the density of SNP is minimum around TAD boundaries, but the authors show that chromosomal rearrangements are enriched at TAD boundaries (in LINE 438-439). The conclusions and the results seem to contradict each other.
14. Why did the authors use TADs identified by TopDom for earlier analysis (in **LINE 271**) but use TADs identified by Arrowhead for the analysis of tissue conservation and specificity (in **LINE 508**) ?
15. This Abstract is too long, and should be simplified.

Reviewer #3:

Remarks to the Author:

Liao et al. described a comprehensive study for 3D structure of a large plant genome using the high quality pepper genome construction. They generated Hi-C data using 4 different tissues and annotated TADs in each tissue using Hi-C data. Finally, they compared TADs and found biological characteristics for epigenomic structures of peppers with distinct features comparing to animals. Overall, I found a great efforts of the authors for this study and new significances and novel insights. Because TADs-related analyses are rare in plant sciences, I think that this study could be one of a guide to understand chromatin structures and the role of TADs for transcriptional regulation. To improve manuscript, I have several concerns with comments.

i) Validation of chromosome structures of pepper genome

Recently, chromosome construction of genome assembly using Hi-C has been an general approach. There were many studies reporting a necessity of validation that can correct anchoring errors from automatic process using Hi-C. Although the authors compared the pepper genome to Zunla genome, it can just reveal only tendency. Because the authors generated Hi-C data in different tissues, it will be helpful for users if the authors compare four sets of pepper chromosomes using Hi-C data in each tissue separately. If the authors detect different chromosome orders among four chromosome sets from Hi-C in four tissues, the authors should evaluate them.

ii) Unbiased annotation of TADs in pepper genome

TADs is a key concept in this study and thus accurate and logical prediction of TADs is a must. Although the authors used 3 tools and integrated those results (1911 TADs), I feel that this is not enough. Because genome structure is much more complex and size is larger than Drosophila, I think this 1911 can be underestimated which is similar to Drosophila, and this is might be because of Arrowhead which generated 1780 TADs. Additional TADs (if the authors more annotate) can contribute to find new biological insights related to 3D structure of large plant genome. Considering other genomic evidences such as clear CTCF and cohesion sites, I think authors can add more TADs in results from three tools.

iii) Simplify for easy understanding

This manuscript contains methodological and biological descriptions and these prevent clear understanding for readers. In particular, I found many of tool-related descriptions. These can be moved to Method section or Supplementary notes. Besides, the description of manuscript is too complicated to understand because there were many of stories with data. I think several things can be moved to Supplementary Notes.

Reviewer #4:

Remarks to the Author:

The authors performed an in-depth analysis of pepper 3D genome architecture: Their state-of-the-art analyses included manifold facets of 3D genome folding, including its relationship to transcriptional activity and evolutionary processes. This is a very nice manuscript, the data is of high quality, and the computational analyses are very well performed. I especially appreciated how the authors are critical about algorithms used and, thus, tried to back up their initial results by using different programs/parameters. Additionally, the authors never overstate their findings and provide meaningful explanations in case results cannot be clearly interpreted. I highly recommend this work for publication.

I have a few comments below. The most important one concerns the use of replicates throughout the manuscript. I did not really understand the reasons for picking certain replicates or not.

Other comments can be found below:

Extended Data Fig.4: Could the authors include an explanation what TSD stands for in the figure legend?

Fig.1 and Extended Data Fig.5: Why were not all replicates used to calculate cis decay and long/short range ratios? The authors previously mention to have replicates for all the samples?

Why do figure 1d and Ext. Data Fig.5e not show the same results? They seem to be described exactly the same way, however, the boxplot is different (e.g. Placenta!). Similar to the previous point, could the authors better explain how they dealt with their replicates?

Figure 5g: The similarity between tissues and between replicates cannot really be distinguished, thus the biological relevance of (sub) compartment changes are likely insignificant. Maybe the authors could stress this more in the text.

p. 12 line 298: This is a very unexpected result. Intuitively, CD borders and TAD borders should definitely overlap (although a CD could easily host several TADs). Could the author elaborate a bit more on this result and probably adapt figure 4 in order to highlight the non-overlapping borders more?

p.23, line 573: I would use "configuration" instead of "conformation".

Reviewer #1 (Remarks to the Author):

In this paper, Liao et al have performed extensive HiC on pepper and have written a report on their findings. Their main findings is that TADs are prominent in pepper, oppositely to other plant species. They also find (sub)compartments that correlate with transcription and compare the 3D organisation between different tissues finding some differences. Interestingly, their most prominent TADs are the ones that are enriched in transposons. They also claim that loops are present at the TAD boundaries and that they may be caused by loop extrusion. Finally, they relate their findings to synteny and conserved sequences finding that breaks of synteny appear at TAD boundaries.

Overall, I think this is a careful and well-executed work that has required a lot of effort. I appreciate the fact that the authors give a lot of details on their experimental and computational analysis and have done the bioinformatics using different complementary tools. At the same time, I also feel that some of their results may be due to misinterpretation of their data. I think before publication in Nat Comm, the authors should significantly revise the paper and strengthen (or drop) some of their claims.

[R1C0]: *We thank the referee for the careful and insightful review of our manuscript. We respond to your comments point-by-point below.*

Comments to address:

1. I typically would not recommend to add more work to a paper that contains already a lot, but this paper clearly lacks some chip-seq. The inactive TADs made up by transposons sound to me like heterochromatic regions that are dense and collapsed. (The absence of gene expression is somewhat already confirming this hypothesis). This picture would thus be very different from the typical TAD in, e.g. humans, where there may not be a single clear epigenetic mark (as that would be considered a compartment).

[R1C1]: *We thank the reviewer for this frank assessment. To address the reviewer's concerns, we generated ChIP-seq data for three histone modifications, including active chromatin mark H3K4me3, heterochromatin mark H3K9me2, and repressive mark H3K27me3, as well as DNA methylation data from the leaf (see **Supplementary Table S7**). Integrating Hi-C maps and TAD-like structures with these new data have made it clear that most of those TAD-like domains annotated in the pepper genome largely correspond to heterochromatin-driven folding domains (i.e. enriched with H3K9me2 mark and retrotransposons), which is consistent with the reviewer's interpretation. To distinguish these chromatin*

domains from those canonical TADs reported in mammals, we call these domains chromatin folding domains or TAD-like domains in the revised manuscript.

1a. Given the point above, it is not clear to me why TADs and compartments do not overlap as most of the inactive transposon-rich TADs should perfectly coincide with B compartments. Can the authors comment on this?

[R1C2]: *We apologize for not making clearer how we compared the overlap between TADs and compartments. As suggested by the reviewer, we indeed find that most (89%) of the TAD-like domains (heterochromatin-driven domains) belong to inactive 'B' compartments (see changes on Page 11).*

In our original analysis, we underestimated the overlap of the TADs and compartments, for a few reasons. First, we considered TAD/compartment overlap by requiring at least 80% reciprocal overlap of their coordinates, which may have been too strict. Second, we compared across analysis methods and bin sizes, which biased our overlap estimates downward.

In the revised version of the manuscript, we compared TADs (HiCExplorer or TopDom) and compartments (Calder) that were called using Hi-C maps at 10-kb resolution and found they identified a comparable number of domains. Calder inferred 13,310 compartment domains, HiCExplorer called 7,829 TADs, and TopDom 14,676 TADs. We found that 36-45% of their boundaries overlapped, and ~18% of their domain bodies exhibit virtually complete overlap (see Fig 4c). These percentages are significantly higher than random expectations. Additionally, for a rough inspection of compartments and TADs, we found that they indeed largely coincide with each other at large ranges (see Fig 2a). Given these results, we now have corrected our previous claim that "TADs are not compartments" to "a large fraction of chromatin folding domains in the pepper genome are compartment domains". (see changes on Page 11).

1b. It looks to me that most of the discussion evolves around the interpretation of HiC maps which is done using different tools, over different tissues or related species. Interpreting HiC maps is notoriously difficult and I don't blame the authors for elaborating at length on this but I have the impression that they sometimes misinterpret the maps (see point 2) and/or they "see too much into them".

[R1C3]: *We thank the referee for carefully reading our manuscript. In the revised manuscript, we substantially revised the "Discussion" and removed many of our previous claims that derived from a possible misinterpretation of our observations (see changes on Page 21-24). We hope the reviewer is satisfied with our changes.*

1c. Related to the point above, I don't understand what's the point in classifying the compartments in 8 subcompartments. Why not 6 or 4? Why not leaving 2 (A/B) compartments? What do we learn from this further subclassification? My interpretation of Figure 2f is that this classification is an overfitting of the PCA.

[R1C4]: *We apologize for not making it clearer why we divide compartments into multiple subcompartments. We were motivated by previous observations that there are multiple*

subcompartments that are associated with different histone modifications in humans (Rao et al 2014). In the revised manuscript, we do not stipulate how subcompartments should be divided. Instead, we aim to identify those important biological features that are associated with subcompartments (e.g. 4 or 8 subcompartments). Based on their genomic and epigenomic features, we found that the subcompartment ranks (i.e. A1.1, A1.2, A2.1, ..., B2.1, B2.2 for 8 subcompartments) are either negatively or positively correlated with a number of genomic (i.e. gene and TE content, and expression level) and epigenomic features (histone modification marks and DNA methylation) (See Fig 2d and Extended Data Fig. 5) We propose that these subcompartments may reflect subtle changes in patterns of epigenomic features, as shown previously (Liu et al., 2021). This is consistent with the claim (Liu et al., 2021) that genomic regions with similar genetic or epigenetic features are more likely to contact each other, or genomic regions in contact with each other are prone to have similar genomic and epigenomic features (see changes on Page 7).

Rao, S. S. P. et al. A 3D map of the human genome at kilobase resolution reveals principles of chromatin looping. *Cell* 159, 1665–1680 (2014).

Liu, Y. et al. Systematic inference and comparison of multi-scale chromatin sub-compartments connects spatial organization to cell phenotypes. *Nat. Commun.* 12, 2439 (2021).

1d. The authors mention that "However, this method failed to resolve the identification of A/B compartments using Hi-C contact matrices at higher resolution (e.g. 40-kb)." Why is this A/B classification failing? Maybe there is some interesting biology/biophysics hidden behind this failure. After all, standard A/B PCA works well in high resolution data (e.g. human HiC at 1kb resolution).

[R1C5]: We thank the reviewer for raising this. We employed several tools (e.g. HiCexplorer, Juicer, FitHiC, FAN-C, etc..) that are based on the standard A/B PCA method to identify the A/B compartments with Hi-C maps at 40-kb resolution. However, we found that these tools do not consistently identify the A/B compartments in our sample. Two common issues may cause this: (1) the first principal component does not always correspond to the A/B compartment. Instead, it may reflect the separation of two chromosome arms or other covariates which explains a large proportion of the variation. This is a common issue of these approaches. Analogous problems occur for other dimensionality reduction analyses when the variance explained by specific principal components cannot be reliably associated with specific mechanisms a priori; (2) related to (1), the methods work for some chromosomes but not others, and work for some samples but not others. Unfortunately, we could not find a satisfying and general solution to the problem. Practically speaking, however, we found Calder worked well for all samples and chromosomes. Additional work may tease out potentially interesting biology/biophysics behind these apparent contradictions, although we're currently uncertain how to go about chasing them down. short of manually assigning each phenomenon to each principal component in each study through direct inspection of the PCs for every chromosome.

1e. I find rather strange that 50% of the genome is in the A (transcriptionally active) compartment (line 124). Given the large amount of transposons I would have expected most of the genome to be in the B (silenced) compartment.

[R1C6]: We thank the reviewer for raising this concern. In the revised manuscript, we used Calder to identify A/B compartments using 10-kb resolution Hi-C maps. We find that with this resolution map, the

coverage of the B compartments has increased to between 59% to 65% across tissues (see Fig 2c), compared to between 48% to 55% we reported previously using 40-kb HiC maps (see Extended Data Fig 4b). We reason that this is because higher resolutions (here 10-kb) recover more refined regional compartments. That is to say, the 40-kb Hi-C maps called more local regions to be A compartments. As a concrete example, consider a 40-kb genomic bin that contains a gene. If the gene causes the bin to be identified as an A compartment, this affects the entire 40-kb region, not just the gene. However, if the 40-kb is subdivided into 4 10-kb bins, the bin containing the gene will be identified as an A compartment, whereas bins lacking genes may be identified as belonging to the B compartment. This finding also reminds us that parameter choice is crucial for bioinformatic analysis. (See changes on Page 7)

2. TAD loops - the authors claim that they observe loops in their HiC maps. This is true, but these loops are clearly very different from the ones observed in humans by Rao et al or other works. First, these loops are *outside* the TAD and coincide with horizontal and vertical stripes that are "depleted" in contacts. On the contrary, the dots on mammalian HiC are *inside* the TADs and coincide with stripes that are "enriched" in contacts. Additionally, the mammalian dots do not coincide with full genes and instead match with CTCF binding sites. (This connects to my previous comment, i.e. lack of chipseq in this work.) As shown in Fig.5e, it is clear that in pepper, these spots in the HiC map are clusters of genes. Since the genes are short, they appear as "spots" or "loops".

[R1C7]: We thank the reviewer for these helpful comments. We agree with the reviewer's comments that the TAD loops we called in the pepper genome are distinct from those in humans. Indeed, we have borrowed the reviewer's framing of this important observation in the revision of the manuscript. To our knowledge, no homologous architectural proteins (e.g. CTCF) have yet been identified in plants. Our newly generated three ChIP-seq datasets and DNA methylation data for the pepper do not change this assessment. Thus, we have no evidence that mechanisms analogous to loop extrusion are supported by our data. Consequently, we removed all discussions pertaining to that mechanism. (See changes on Page 14)

2a. A simple model to explain these "loops" of genes is the transcription factory model by Peter Cook and coauthors or the transcription hubs/clusters (or even condensates) models (See I Cisse, R Young, etc). In these models transcription factors form bridges between different genes and glue them together in hubs. No need of loop extrusion to explain these HiC features. See <https://academic.oup.com/nar/article/44/8/3503/2467087>
<https://www.nature.com/articles/s41467-021-25875-y>

[R1C8]: We thank the reviewer to help explain the loops we observe in the pepper genome! We agree with the reviewer and have now highlighted these models to explain our observations. We also cited the pertinent references in our manuscript (see changes on Page 14).

2b. From Fig.5e is also clear that the PC analysis and TAD calling struggle to capture these small islands of active genes at this resolution. They are identified as TAD boundaries but are effectively small compartments upon closer visual examination.

[R1C9]: Yes, we have confirmed that these TAD-like domain boundaries are enriched for genes that are identified as small subcompartments by Calder, especially when we used Hi-C maps at 10-kb resolution (see updated Fig. 5e).

2c. I strongly suggest the authors to revise their text around the statements of TAD loops and loop extrusion and/or provide stronger evidence for a mechanism (loop extrusion or others) for these features. For instance "predominantly partitioned into TADs and that a large proportion of TADs are likely formed by loop extrusion..." needs to be removed as it is likely wrong. SMC chipseq data is otherwise needed to support this statement.

[R1C10]: We thank the reviewer for this suggestion. We agree with the reviewer's comments. We have now revised our claims that 'loop extrusion' is the major formation mechanism of TADs in the pepper genome. Following the reviewer's suggestion and our new results drawn from the ChiP-seq data, we now conclude that the TAD-like domains identified in the pepper genome are likely heterochromatin-driven folding domains that may be promoted by transcription factories. (See changes on Page 14)

3. I am rather impressed by the similarity of small-scale 3D genome organisation (TADs and features near the diagonal) across tissues (Fig.3c). The leaf clearly displays a Rab1-like long-range contacts but apart from this, the small scale arrangement doesn't seem too different for the other tissues. This suggests that TADs may not be needed to control gene expression, as mentioned by the authors. Can the authors probe this further? Are there genes that can be turned on in response to external stimuli such as heat/cold? Can they measure (via HiC, or 4C) the change in organisation upon activation?

[R1C11]: We thank the reviewer for this feedback. The relationship between spatial chromatin features (e.g. compartments, TADs, and loops) and gene expression regulation remains controversial. The observation that genomic regions exhibiting variation in chromatin conformation are not enriched for differentially expressed genes (Supplementary Table S15) suggests that chromatin conformation doesn't predict differential gene expression between tissues. However, our results do suggest that chromatin conformation is associated with modulation of existing differential expression (Fig. 7), perhaps by facilitating the action of existing regulatory elements (see below References).

Thus, we predict that variation in chromatin conformation (at least for compartments, chromatin domains, and loops investigated in this study) does not actively establish new patterns of differential expression. Rather, it may modulate existing mechanisms that establish expression variation (i.e. magnitude of expression level). Such differences for the cause-and-effect relation between changes in chromatin conformation and gene expression variation can be measured through Hi-C (Fig. 7). We suggest that further studies need to focus on the quantitative effect of the chromatin conformation in gene expression regulation.

Despang, A. et al. Functional dissection of the Sox9-Kcnj2 locus identifies nonessential and instructive roles of TAD architecture. Nature Genetics vol. 51 1263–1271 (2019).

Ghavi-Helm, Y. et al. Highly rearranged chromosomes reveal uncoupling between genome topology and gene expression. Nat. Genet. 51, 1272–1282 (2019).

Espinola, S. M. et al. Cis-regulatory chromatin loops arise before TADs and gene activation, and are independent of cell fate during early Drosophila development. Nature Genetics vol. 53 477–486 (2021).

4. Again on TADs/compartments: I disagree on the statement at line 591 "One striking ... more defined TADs". I would like the authors to think about the definition of TAD and that of compartments. At what point a compartment is a TAD and vice versa? Figure 3a is an example of this, the checkerboard pattern on the left half of the panel is a signature of a compartment that can be made by bridge proteins binding specific epigenetic marks (See again <https://academic.oup.com/nar/article/44/8/3503/2467087>). Yet, the squares near the diagonal will be wrongly classified as TADs. I suggest the authors to critically think about their results - why should a larger genome size imply the existence of TADs? Is it not more likely that Pepper displays (in accordance of other plants) a lack of TADs and that the ones seen in this work are instead wrongly classified? Indeed, by staring at extended data Fig.1, it is not clear to me that pepper HiC map is different from the rice one, apart from that the rice displays smaller domains (but it is also true that the scale is different in the map ~3Mbp VS ~20Mbp).

[R1C12]: Motivated by the reviewer's comments, for which we are very grateful, we have substantially revised our manuscript and removed these statements. Based on our new ChIP-seq data, especially the heterochromatin mark H3K9me2, it is now clear that most TAD-like domains identified in the pepper genomes are heterochromatin-driven folding domains, probably due to heterochromatin collapse of silenced regions (see Fig. 4), as the reviewer suggested. Throughout this manuscript, we now use TAD-like domains or self-interaction domains, or chromatin domains to distinguish our observations from canonical TADs described in mammals. Indeed, we found that approximately 89% of these heterochromatin domains are overlapping with the 'B' compartments. Regarding why larger plant genomes display more prominent TAD-like domains, we reasoned this is because larger genomes have more TEs than small plant genomes, which is consistent with previous observations that TAD-like domains are more likely displayed at larger plant genomes and pericentromeric heterochromatin regions (see below references).

Liu, C., Cheng, Y.-J., Wang, J.-W. & Weigel, D. Prominent topologically associated domains differentiate global chromatin packing in rice from Arabidopsis. Nat Plants 3, 742–748 (2017).

Doğan, E. S. & Liu, C. Three-dimensional chromatin packing and positioning of plant genomes. Nat Plants 4, 521–529 (2018).

4a. The potential pitfall in highlighting the presence of TADs in this work is also clearly seen in the fact that 3 different tools yield substantially different TAD annotations (extended data fig 7). Which one should we trust?

[R1C13]: We thank the reviewer for this comment. The inconsistency of TAD annotation among tools seems to be common both in benchmark analyses (see below references) and in practice. For example, in our previous work on Drosophila, we also noticed a substantial difference in TAD annotations. In that work, we collected more than 700x coverage of Hi-C data (Liao et al, 2021). We think the reason for this difference is at least partially caused by the different algorithms adopted in these methods. In an attempt to alleviate this effect, we thus conducted analyses for TAD-like domains identified by multiple methods that consider different features of Hi-C data. More frequently, we used

the TAD-like domain annotated by TopDom in our analyses due to its top performance among all tools as shown in a previous benchmarking work.

Forcato, M. et al. Comparison of computational methods for Hi-C data analysis. Nat. Methods 14, 679–685 (2017).

Zufferey, M., Tavernari, D., Oricchio, E. & Ciriello, G. Comparison of computational methods for the identification of topologically associating domains. Genome Biol. 19, 217 (2018).

4b. On a global level, it may seem that pepper has more longer range contacts than the other plants. Can the authors plot the contact probability of pepper against that of the other plants? What's the difference in volume fraction occupied by the genome in the nuclei of these species? Is pepper's genome much more crowded/confined than the genome of other plants?

[R1C14]: We thank the reviewer for this suggestion. We plotted the frequency for long versus short contacts in rice, maize, tomato, soybean, and pepper (see below). It should be noted that we are not sure whether they are comparable to each other because the Hi-C data were collected in different tissues, times, and batches, and such differences may influence the frequency, thus we did not include this plot in our manuscript. Generally, large genomes have a higher ratio of long-versus-short contact frequency. We think that comparisons between species need to be more carefully considered. We apologize that we have no such knowledge about the volume fraction occupied by the genome in the nuclei in plant species. Due to this unclear information, we have removed this kind of discussion in our manuscript (See changes on Page 21).

5. I don't understand how can TADs not coincide with a subcompartments (line 616, "only 11% of TADs coincide with ..."). From what I understand, each genome segment is assigned to a (sub)compartment, so by definition each TAD should contain one or multiple (sub)compartments. Can the authors clarify?

[R1C15]: We apologize again for not making clear how we compared TADs and compartments. We considered that a TAD domain overlapped with a compartment domain when their coordinates exhibited

at least 80% reciprocal overlap of length. For our analyses, you can refer to our response to **R1C2**. You can also see an example from **Fig. 4f**, how we align the TADs and compartments.

Practically, each genome segment is assigned to a (sub)compartment (such as a 40-kb bin), for example, A1.1, and if its flanking bins were also assigned as the same subcompartment rank A1.1, these three 40-kb bins will jointly be combined as a single subcompartment. By this approach, a compartment can be similar in size to TADs, especially when we annotated both using Hi-C maps at 10-kb resolution. They can be fully or partially overlapping.

6. Line 618. Another example of misleading interpretation of the evidence: "Based on these ... HiC maps". First, these HiC maps show no evidence of loop extrusion (the spots are not inside TADs but are made by neighboring active genes that are clustered). Second, a simpler explanation is the heterochromatic collapse of silenced regions.

[R1C16]: *We agree with the reviewer's comments, and again we are very grateful for them! Accordingly, we substantially revised and corrected the interpretation of our observations related to "loop extrusion" throughout the manuscript. (see changes on Page 14).*

Reviewer 2 (Remarks to the Author):

In this manuscript, the authors assembled a *Capsicum annuum* genome and constructed Hi-C maps for four tissues, including leaf, bud, pulp and placenta. The title of the manuscript is “The 3D architecture of the pepper (*Capsicum annuum*), genome and its relationship to function and evolution”. However, I did not find any specific analyses related to function, and the analysis of evolution is only reflected in a section “Breaks of synteny preferentially occur near TAD boundaries, despite high evolutionary”. Overall, many parts of this manuscript gave only some descriptive results from data analysis, without functional study of at least one chromatin structure with specific functional implications. The authors obtained some general conclusions about 3D genome organization in pepper, but most of them were previously known in other species.

[R2C0] Thank you for your careful reading of our manuscript and critical remarks. While we might appear to differ in terms of what we find functionally relevant (we believe both gene expression and chromatin conformation are important aspects of genome function), we nevertheless share the reviewer’s strong desire for even more functional data. Consequently, we’re happy to introduce additional functional experiments, collect new data, and conduct new analyses. We’re also happy to receive the feedback of the reviewer, as it helps us hone our interpretation of the results.

Thanks in part to the reviewer’s comments, we believe our revised manuscript now provides ample functional insights. Our work represents advances in three areas: 1) functional genomics; 2) genome evolution; 3) genomics resources. In particular, this work represents a thorough treatment of an important cellular phenotype: the 3-dimensional shape of chromosomes in the nucleus. Integration of this fundamental trait with gene expression, methylation, and chromatin state helps us understand the relationships between chromosome shape and important molecular phenotypes, like transcriptional activation and repression, among others. Additionally, our results advance our understanding of the rules of genome evolution in relation to chromosome shape and are some of the first such observations in any plant. In particular, we provide strong evidence that genetic variations including SNPs, SV, and chromosome rearrangements are associated with underlying chromatin conformation. Finally, we provide a set of genome resources of unprecedented quality for the pepper community. Indeed, our work compares favorably to any plant assembly in terms of continuity, completeness, accuracy, and amount of functional data for annotation. While our research goals for this project were not aimed at individual locus-level functional dissection, our work does provide global functional genomics information at a scale and resolution only made possible by recent advances in genomics techniques. We believe our work to be commensurate with the scope and topics of other works of genomics in Nature Communications.

While we agree with the reviewers that some of these observations have been shown in other organisms, at this stage in the development of the field it remains important to identify patterns on relevant biological axes, like placement on the phylogenetic tree, genome size, repeat content, etc. The pepper assembly is ~3Gb, making it not only one of the larger and more repetitive reference genome assemblies in Dicots, but across most plants. It’s certainly an interesting example to compare to other closely related assemblies, which have to date typically been smaller than 1Gb. Indeed, studies like ours permit the field to generalize about the types of variation we expect and to calibrate our sense of how quickly conformational differences accumulate as one traverses the tree of life. Finally, we have made some novel observations. In particular, despite the reputation of plant genomes for malleability (perhaps influenced by studies of maize genomics), there are strong patterns of evolutionary constraint

that vary with their positions relative to the conformational domains we've identified here across a broad array of genetic changes. To our knowledge, this is the first such observation in plants. We hope that you agree that our revisions improved the manuscript. We also carefully considered and responded to each of your comments below.

My other comments are shown below:

1. Please explain “Phred Quality Score of Q52” in line 128 and “EBP 6.C.Q40 standard” in Line 144

[R2C1]: We apologize for lacking clarity. We have now included a paragraph in the Method section and cited the original reference to describe how the Phred Quality Score and EBP 6.C.Q40 standard were calculated (see Changes on Page 4 and 25).

2. The intact LTR-RTs have two peaks (~0.2 Ma and ~2 Ma) in the pepper genome. Are there additional peaks if truncated LTRs are included for analysis? Because 2% of repeats cannot represent the LTR burst time of the entire genome.

[R2C2]: Motivated by the reviewer's comment, we performed an additional analysis for all LTR sequences based on sequence divergence between pairwise alignments (see below, reported in Supplementary Fig. S4). Although this is a preliminary analysis, we can still glean important features about the tempo and timing of LTR evolution in the pepper genome. Notably, we identified at least 4 peaks of LTR divergence in the pepper genome, suggesting the presence of 4 bursts of TE activity in the past. This result is now presented in **Supplementary Note 2**.

3. Previously published genome and the current assembly should be compared. This may provide some evidence showing the high quality and improvement in the current assembly.

[R2C3]: Thank you for this suggestion. We have compared the basic assembly quality metrics (such as Contig N50, BUSCO, and so on) in **Supplementary Table S3** from 6 previous pepper assemblies. We also compared the intact LTR elements identified in previous pepper assemblies (**Supplementary**

Table S4). All these comparisons support our assembly being significantly improved at both continuity and completeness levels.

4. References in Line 97-98 showing “recent studies in wheat have reported that chromatin loops demarcate a large proportion of chromatin domains” should be added.

[R2C4]: We apologize for this oversight. We now have included the reference in the revision, and carefully checked throughout the manuscript for possible other missing references.

5. Did the authors performed a sequencing depth analysis for Hi-C data in different tissues? Different depths regarding of the number of valid interactions have an impact on the comparison between different tissues.

[R2C5]: Thank you for this comment. It is indeed important to reassure readers that our comparisons are reliable and not subject to the sorts of biases described by the reviewer. To this end, we have reported quality metrics for our data across samples in **Supplementary Table S9** (conducted by Juicer) and **Supplementary Table S8** (conducted by HiCExplorer). We also reported the map resolutions and data consistency across tissues in **Supplementary Table S10** and **Supplementary Table S12**, respectively. Generally, we generated 60-80-fold coverage of nucleotide coverage from the Hi-C reads for each sample, with between 335M and 487M valid pairs for each sample. Given how close the total sequencing depth is between samples, variation in coverage is unlikely to influence our inferences. Indeed, the different tissues display remarkable similarities, which would further support this assessment.

6. Why not show corrected Hi-C counts from 0~106 in Fig. 1c. The unit of the x-axis should be added.

[R2C6]: Thank you for pointing this out. We now include this in the revised manuscript.

7. Why did the authors use CPM instead of FPKM or TPM to represent the expression levels of genes? The CPM value is correlated with the gene length.

[R2C7]: Thank you for this comment. If we're interpreting this comment correctly, we think the reviewer is referring to Fig. 4f. We used CPM to represent the expression level because the CPM values were calculated for each 10-kb or 40-kb bin rather than for each gene. Unlike for genes, the unit of analysis is constant in length, so no length dependency can arise. However, this is a common response to this analysis. As a result, we now point out this feature to make our intent more clear. When we did perform gene-level analyses, we used TPM calculated for each gene to represent the expression level (see **Extended data Fig. 5b**). However, specifically for differential gene expression analysis (**Supplementary Table S15 and Fig. 7**), we used CPM, as it is the standard input for the LIMMA R package.

8. Abbreviations of the units that indicate chromatin length in figures include Mpb (Fig. 1b, Fig. 2f, Fig. 3c, Fig. 3d, Fig. 4a, Fig. 4f), Mb (Fig. 3b, Fig. 4c, Fig. 5a, Fig. 5e), kbp (Fig. 5d), kb (Fig. 4h, Fig. 2d), Kb (Fig. 6e) and none (Fig. 1c, Fig. 3a). It is better to be consistent.

[R2C8]: Thank you very much for your careful review. We correct the Abbreviations throughout the manuscript, and now they are consistent.

9. Some TADs are not conserved/consistent in four tissues and some are not prominent in Fig. 3c.

[R2C9]: Thank you for raising this. The annotated TADs across tissues are somewhat different but are largely preserved between tissues. We reasoned this difference may have partially resulted from tissue-specific features and/or technical biases. However, subtle tissue-specific differences may or may not derive from factors directly related to chromatin conformation in the nucleus. We have now corrected our statements to communicate the nuances more clearly. We now mention that TAD-like domains do exhibit some variation between tissues. We have also deleted the word “prominent”, instead noting that the TAD-like domains occupy a substantial proportion of the genome, which we believe remains accurate. With these clarifications, we hope the text of the manuscript now conforms better to the reviewer’s interpretation of the results. (See changes on Page 10).

10. “In all comparisons, we found less than 11% of TADs coincide with CDs. This result suggests that compartmentalization may contribute to the formation of only a small fraction of TADs in the pepper genome” in LINE 296-299. I believe that the overlap between two structures is related to the size of the structure. If the authors characterize the overlap ratio between TADs and compartment, this value may be higher than 11%. I suggest that more details should be shown to support the conclusion that compartmentalization may contribute to the formation of only a small fraction of TADs in the pepper genome.

[R2C10]: We thank the reviewer for this helpful suggestion. Consistent with the reviewer’s hypothesis, the overlap level is indeed affected by the size of compartments and TAD-like domains. The unequal number of both domains made us underestimate the overlap of the compartment and TADs. When we used the HiC map at 10-kb resolution, Calder identified 13,310 domains and TopDom identified 14,676 domains, these similar numbers permitted a higher overlap ratio (~18%), and the boundaries overlap with ~36-45%. Thus, in the revised manuscript, we corrected our previous claim, and instead, we suggest that a substantial fraction of TAD-like domains are compartments. (See changes on Page 11)

11. Using a 40-kb resolution leaf Hi-C map, these three methods identified 1,780, 4,663, and 2,641 TADs in LINE 264-265, but the sum of TADs in Fig. 4b is 2640.

[R2C11]: Thank you very much for pointing this out. We miscounted the number of the TADs. We have now corrected it in the revised manuscript!

12. The authors mention that “the most prominent TADs formed in genomic intervals with a long stretch of enriched LTR-RTs and were always flanked by active transcription regions (Fig. 4f and Extended Data Fig. 8d)” in LINE 319-321. It is not enough to use two examples to illustrate this phenomenon and more statistics are needed to support it.

[R2C12]: *We agree with this comment. While such examples are easy to identify in the plots, quantifying this observation is difficult due to our vague phrasing used in the initial manuscript draft (particularly because we didn't quantify what we meant by "most prominent"). So, rather than try to salvage that statement, we removed it in favor of a more comprehensive and quantitative description of the chromatin state in light of the new data we collected. In the revised manuscript, we comprehensively compare TAD-like domains in each group using a number of genomic (Gene and TE content) and epigenomic features (ChIP-seq from H3K4me3, H3K27me3, and H3K9me2, together with DNA methylation data). We found that approximately 60% of the genome is covered by heterochromatin and TE-rich domains. We removed the statement above and substantially revised the manuscript and corrected this claim. (See changes on Page 11)*

13. Fig. 6c shows the density of SNP is minimum around TAD boundaries, but the authors show that chromosomal rearrangements are enriched at TAD boundaries (in LINE 438-439). The conclusions and the results seem to contradict each other.

[R2C13]: *We thank the reviewer for raising this. We carefully confirmed the results by checking the pipelines that we used to analyze the data. Although these two conclusions seem contradictory to each other, they may reflect different constraints exerted by chromatin conformation or genic conservation. First, genetic variants, such as SNP and InDels, that are depleted at TAD-like domain boundaries may be due to genic conservation at such regions, consistent with the enrichment of genes. But, we can't rule out other confounding effects resulting from functional or structural elements that confer a beneficial 3D conformation present at the boundaries. Second, chromosomal rearrangements tend to maintain the integrity of TAD-like domains, likely due to these domains being involved with functional spatial structures, such as transcription factories. Similar patterns have also been shown in mammals and Drosophila (see below references), but not yet in plants, so there is precedence for this interpretation. (see the explanation on Page 23)*

Krefting, J., Andrade-Navarro, M. A. & Ibn-Salem, J. Evolutionary stability of topologically associating domains is associated with conserved gene regulation. BMC Biol. 16, 87 (2018).

Fudenberg, G. & Pollard, K. S. Chromatin features constrain structural variation across evolutionary timescales. Proc. Natl. Acad. Sci. U. S. A. 116, 2175–2180 (2019).

Liao, Y., Zhang, X., Chakraborty, M. & Emerson, J. J. Topologically associating domains and their role in the evolution of genome structure and function in Drosophila. Genome Res. 31, 397–410 (2021).

14. Why did the authors use TADs identified by TopDom for earlier analysis (in LINE 271) but use TADs identified by Arrowhead for the analysis of tissue conservation and specificity (in LINE 508)?

[R2C14]: *Thank you for pointing this out. We performed analyses for TAD-like domains annotated by both Arrowhead and TopDom. To improve consistency, we have now reported results from TopDom in the main text and moved results from Arrowhead results to **Extended Data Fig. 11**. This doesn't change the conclusions we draw and improves the clarity for the reader.*

15. This Abstract is too long and should be simplified.

[R2C15]: *Thank you. We have now shortened the abstract to less than 150 words.*

Reviewer #3 (Remarks to the Author):

Liao et al. described a comprehensive study for 3D structure of a large plant genome using the high quality pepper genome construction. They generated Hi-C data using 4 different tissues and annotated TADs in each tissue using Hi-C data. Finally, they compared TADs and found biological characteristics for epigenomic structures of peppers with distinct features comparing to animals. Overall, I found a great efforts of the authors for this study and new significances and novel insights. Because TADs-related analyses are rare in plant sciences, I think that this study could be one of a guide to understand chromatin structures and the role of TADs for transcriptional regulation. To improve manuscript, I have several concerns with comments.

[R3C0]: We are grateful to reviewer #3 for their positive assessment of our work. We carefully consider and respond to the reviewer's concerns and comments below.

i) Validation of chromosome structures of pepper genome

Recently, chromosome construction of genome assembly using Hi-C has been an general approach. There were many studies reporting a necessity of validation that can correct anchoring errors from automatic process using Hi-C. Although the authors compared the pepper genome to Zunla genome, it can just reveal only tendency. Because the authors generated Hi-C data in different tissues, it will be helpful for users if the authors compare four sets of pepper chromosomes using Hi-C data in each tissue separately. If the authors detect different chromosome orders among four chromosome sets from Hi-C in four tissues, the authors should evaluate them.

[R3C1]: We thank the reviewer for raising this insightful suggestion. We agree with the reviewer that it is necessary to carefully consider when applying Hi-C data in the ordering and scaffolding of contigs in the assembly process. We indeed found several errors caused by the automatic Hi-C scaffolding process. Thus, we also performed a manual correction step to finalize our assembly (see examples below). Motivated by the reviewer's suggestion, we also compared the Hi-C maps across the four tissues to evaluate their consensus for genome scaffolding. We found that all Hi-C maps recovered a consensus assembly (see the contact maps on Fig. 1a).

ii) Unbiased annotation of TADs in pepper genome

TADs is a key concept in this study and thus accurate and logical prediction of TADs is a must. Although the authors used 3 tools and integrated those results (1911 TADs), I feel that this is not enough. Because genome structure is much more complex and size is larger than *Drosophila*, I think this 1911 can be underestimated which is similar to *Drosophila*, and this is might be because of Arrowhead which generated 1780 TADs. Additional TADs (if the authors more annotate) can contribute to find new biological insights related to 3D structure of large plant genome. Considering other genomic evidences such as clear CTCF and cohesion sites, I think authors can add more TADs in results from three tools.

[R3C2]: Thank you for raising these concerns and offering these suggestions. We agree with the reviewer that annotation of TAD-like domains with different tools alone is not sufficient to fully resolve the general features of chromatin domains. Thus, in the revised manuscript, we generated additional ChIP-seq data (e.g H3H4me3, H3K27me3, and H3K9me2) and DNA methylation data to further characterize the genomic and epigenomic features of the TAD-like domains in the pepper genome. We now found that a large fraction of TAD-like domains (occupying ~60% of the pepper genome) are heterochromatin-driven folding domains and that these domains are under functional and structural constraints. The addition of this data has permitted us to make direct inferences based on functional genomic information rather than indirect inferences based only on domain annotation and sequence content alone. We think this approach, though expensive and time-consuming, is probably the most informative for identifying biological features related to the 3D structure of this large plant genome. We hope the revisions have addressed the spirit of the referee's suggestions. (see changes on Page 10)

iii) Simplify for easy understanding

This manuscript contains methodological and biological descriptions and these prevent clear understanding for readers. In particular, I found many of tool-related descriptions. These can be moved to Method section or Supplementary notes. Besides, the description of manuscript is too complicated to understand because there were many of stories with data. I think several things can be moved to Supplementary Notes.

[R3C3]: *We apologize for not making the manuscript more understandable. As the reviewer notes, as previously written, the manuscript was quite complicated. We really appreciate the referee taking the time to review our work with fresh eyes. Consequently, in order to improve the readability of the manuscript, we have made substantial changes: 1) As suggested by the reviewers, we have removed some parts with methods described to the Supplementary Results or Methods, such as “The relationship between compartments and gene expression” section; 2) we have cut ~17% of the content for the manuscript, especially for the Results and Discussion sections, compared to our original version to make the manuscript more concise; 3) We polished the manuscript throughout to improve its fluency and logical flow. We hope the changes have improved the manuscript’s clarity.*

Reviewer #4 (Remarks to the Author):

The authors performed an in-depth analysis of pepper 3D genome architecture: Their state-of-the-art analyses included manifold facets of 3D genome folding, including its relationship to transcriptional activity and evolutionary processes. This is a very nice manuscript, the data is of high quality, and the computational analyses are very well performed. I especially appreciated how the authors are critical about algorithms used and, thus, tried to back up their initial results by using different programs/parameters. Additionally, the authors never overstate their findings and provide meaningful explanations in case results cannot be clearly interpreted. I highly recommend this work for publication. I have a few comments below. The most important one concerns the use of replicates throughout the manuscript. I did not really understand the reasons for picking certain replicates or not.

[R4C0]: *Thank you for your feedback and enthusiasm.*

First, we would like to introduce the details for how we deal with the replicates in our data analyses, because it seems to be the biggest concern of the reviewer. We generated Hi-C data for four tissues, each with two replicates. But we collected the Hi-C data at two different times, which means the samples represent two batches.

For chromatin features, like compartments and TAD-like domains, we called them in all eight samples separately. We treated all samples as independent, and our results showed that these features are highly conserved between samples. Thus, our results just provided a qualitative description of these features, rather than showing how the difference between samples.

For loops, we combined each replicate to get a high coverage of Hi-C data for its annotation. But we highlight that “Importantly when loops detected in one tissue were missing in another, we could not exclude the possibility that they were present but below the threshold of detection. We reasoned that this might be due to technical limitations in loop detection approaches or reflect subtle changes in the interaction frequency between tissues.”

For many of our results, we performed quantitative comparisons, particularly in assessing associations between conformation and other biological features. Replication certainly helps these comparisons, as the reduction in technical and sampling noise that comes with replication makes these associations more reliable and easier to observe. Fortunately, for our study, these patterns are sufficiently strong that they are recoverable with only modest replication. In the last section (ie “the relationship between chromatin features and gene expression”) we considered replication more extensively. For those analyses, we adopted the following approaches: 1) we evaluated the consistency of results between independent analyses done for each replicate; or 2) used only the chromatin features shared by both replicates for analyses. These analyses led to a similar conclusion. We added a description of these strategies in the Method section (see changes on Page 31).

We respond to your comments point-by-point below.

Other comments can be found below:

Extended Data Fig.4: Could the authors include an explanation what TSD stands for in the figure legend?

[R4C1]: *We apologize for this lack of clarity. 'TSD' refers to Target-Site Duplication. We have included this in the figure legend. To be noted, we have moved the Extended Data Fig. 4 to Supplementary Fig. S4 in the revised manuscript.*

Fig.1 and Extended Data Fig.5: Why were not all replicates used to calculate cis decay and long/short range ratios? The authors previously mention to have replicates for all the samples?

[R4C2]: *Our samples are taken from two batches--i.e. because we collected the Hi-C data at two different times. We do the calculation of cis decay and long/short-range ratios separately to reduce the influence of the batch effect. We reported the result (HiCExplorer) of one batch in the main text (Fig. 1c,d) and the other in Extended Data Fig.3c. The results are consistent with each other. We also reported the result derived from Juicer in Extended Data Fig. 3d,e, for samples in one of the batches. We already confirmed the consistency of results between samples in these two batches using HiCExplorer contact maps. To avoid reporting too many redundant results, we did not include the Juicer contact maps from another batch, though we indeed found that the results are consistent. Generally, for the qualitative measures of chromatin features, like, compartments and TAD-like domains, we performed analysis for all eight samples. We combined the Hi-C data from replicates only for loop detection, to increase the Hi-C data coverage which is especially important for loop detection.*

Why do figure 1d and Ext. Data Fig.5e not show the same results? They seem to be described exactly the same way, however, the boxplot is different (e.g. Placenta!). Similar to the previous point, could the authors better explain how they dealt with their replicates?

[R4C3]: *In Figure 1d, we reported the Hi-C data that was processed through HiCExplorer, while in Extended Data Fig.3e (we have changed Fig. 5 to Fig 3 in the revised manuscript), the Hi-C matrices were processed by Juicer. These two Figures are not comparing replicates, but rather comparing different methods. So, the results are slightly different, likely because the two pipelines recovered vary in how they treat short-range contacts, as seen in Supplementary Table S8 (for HiCExplorer) and Supplementary Table S9 (for Juicer). Generally, juicer reported between 11% and 24% contacts as short contacts (<20 Kb), whereas HiCExplorer reported between 2.8% and 5.6% as short contacts. However, both methods recover a similar percentage for long contacts and inter-chromosomal contacts. Thus, this difference is caused by different methods rather than by replicates.*

Figure 5g: The similarity between tissues and between replicates cannot really be distinguished, thus the biological relevance of (sub) compartment changes are likely insignificant. Maybe the authors could stress this more in the text.

[R4C4]: *We thank the reviewer for helping us articulate this point. This has helped us make a few important observations clearer in the manuscript. First, as the reviewer notes, subcompartments don't often exhibit large changes across tissues. However, we neglected to make clear that our comments regarding subcompartments are most relevant when subcompartments **do** vary. Even so, we agree that subcompartment variation exhibits no clear association with differential gene expression. However, we did establish a few nuanced patterns. For example, shifts to lower compartment rank or higher compartment rank are associated with **modulation** of existing gene expression differences to lower and higher levels, respectively. Moreover, epigenetic features (like histone modifications and DNA methylation) are also directly associated with compartment rank (though, as the reviewer notes, this need not lead to subcompartment differences between tissues). We think these more subtle patterns are worth noting. To address the reviewer's concerns, we have clarified these observations. We think our text now acknowledges the limitations of associations between subcompartments and molecular function and is now more clear than before.*

p. 12 line 298: This is a very unexpected result. Intuitively, CD borders and TAD borders should definitely overlap (although a CD could easily host several TADs). Could the author elaborate a bit more on this result and probably adapt figure 4 in order to highlight the non-overlapping borders more?

[R4C5]: *We thank the reviewer for pointing this out. We corrected this statement by performing additional analyses. Please see more details in our responses in **R1C2 and R2C10**.*

p.23, line 573: I would use "configuration" instead of "conformation".

[R4C6]: *Thank you for this suggestion. We have replaced the "conformation" with "configuration", accordingly. (see changes on Page 5)*

Reviewers' Comments:

Reviewer #1:

Remarks to the Author:

The authors have addressed my comments.

Reviewer #2:

Remarks to the Author:

The authors have addressed my previous concerns. This manuscript is largely improved. At this stage, I still have two concerns:

1. In the Methods "TAD annotation and classification", the authors have used two approaches to determine the consistency and reproducibility of TADs identified with different methods (HiCFindTADs, TopDom, Arrowhead) and in different tissue samples. I'm wondering why these two approaches were used. Can Jaccard be used to determine the one-to-one TAD coordinates?
2. Fig. 3c still does not convince me that "TAD-like domains are highly conserved across the four studied tissues.", since some TAD domains are very not unclear. I suggest the authors add insulation scores of TAD-like domains, and also try to give some evidence showing whether those structures are significantly conserved or dynamic. I know it is not easy, but I hope to see some analyses about the differences in 3D structures and even tissue specificity.

Reviewer #3:

Remarks to the Author:

I am satisfied for the author's response for the reviewer's concerns. I agree that the current ms is enough for publication in this journal.

Reviewer #4:

Remarks to the Author:

The authors answered all my questions and incorporated all my comments in the final version of the manuscript. I do not see any further need for changes to publish this very nice work.
Well done!

Reviewer #1 (Remarks to the Author):

The authors have addressed my comments.

Thank you.

Reviewer #2 (Remarks to the Author)

The authors have addressed my previous concerns. This manuscript is largely improved. At this stage, I still have two concerns:

We thank the referee for carefully reviewing the new version of the manuscript. These new suggestions help us make the observations in the manuscript more robust and improve its clarity.

1. In the Methods “TAD annotation and classification”, the authors have used two approaches to determine the consistency and reproducibility of TADs identified with different methods (HiCFindTADs, TopDom, Arrowhead) and in different tissue samples. I’m wondering why these two approaches were used. Can Jaccard be used to determine the one-to-one TAD coordinates?

We initially chose these methods because: 1) it allows us to explore different characteristics of TAD-like structures in the pepper genome. Each method has slightly different constraints on the TADs it infers. For example, in HiCExplorer TADs are consecutive, and Arrowhead they are generally more dispersed, while TopDom is intermediate between the previous two, allowing boundary regions between TADs; 2) Based on manual inspection of the Hi-C maps, we found TAD-like domains in pepper are mostly adjacent along the chromosome (see below Hi-C maps for randomly selected regions from Chr05-Chr07, Chr10-Chr12). For this reason, we chose TopDom to perform the analyses. TopDom also performed well in a published benchmarking analysis(Zufferey et al. 2018).

However, as suggested by the reviewer, we also explored the method TADtool (Kruse et al. 2016), which is based on the insulation index (see below). We have incorporated these new analyses into the latest revision of the manuscript [Page 9-10 and Supplementary Note 3].

We apologize for not being explicit in how we used the Jaccard coefficient to determine the consistency and reproducibility of the TAD-like domains across samples. To calculate the Jaccard distance based on TAD characteristics (i.e. genome coverage, TAD number, and TAD boundaries), we used the following formula:

$$J(A, B) = \frac{|A \cap B|}{|A \cup B|},$$

Taking TAD number as an example, where, A is the number of TAD identified in one sample, and B is the number of TAD identified in another sample. We have added a detailed description in Method.

Kruse, Kai, Clemens B. Hug, Benjamín Hernández-Rodríguez, and Juan M. Vaquerizas. 2016. “TADtool: Visual Parameter Identification for TAD-Calling Algorithms.” *Bioinformatics* 32 (20): 3190–92.

Zufferey, Marie, Daniele Tavernari, Elisa Oricchio, and Giovanni Ciriello. 2018. “Comparison of Computational Methods for the Identification of Topologically Associating Domains.” *Genome Biology* 19 (1): 217.

2. Fig. 3c still does not convince me that “TAD-like domains are highly conserved across the four studied tissues.”, since some TAD domains are very not unclear. I suggest the authors add insulation scores of TAD-like domains, and also try to give some evidence showing whether those structures are significantly conserved or dynamic. I know it is not easy, but I hope to see some analyses about the differences in 3D structures and even tissue specificity.

Thank you. Following the reviewer’s suggestion, we added the analysis for TAD annotation inferred from TADtool (which is based on the insulation index) in the revised manuscript. Applying this new method to call TAD-like domains for the pepper genome revealed that ~75% of the pepper genome is covered by TAD-like domains. About 66% of the TADs identified in TADtool were also inferred by other methods. The pattern of the insulation index across the genome agrees with our interpretation that TADs are largely adjacent across chromosomes. [Supplementary Note 3 or see below Figure as convenience]

Additionally, we found that the insulation index suggests the following groupings: ((leaf, bud), (pulp, placenta)). This result is consistent with the hierarchical clustering of the samples based on the similarity of TAD number and boundaries [Supplementary Note 3 or see below Figure as convenience]. This remains consistent with our interpretation in the previous version of the manuscript for TADs inferred by TopDom.

We next consider the conservation and divergence of TAD-like structures between samples. We found that at least 85% of TADs identified in one tissue are found in one or more of the remaining tissues. See results for TopDom in Figure 3e and Tadtool in Supplementary Note 3 (also shown below for convenience). These results suggest that the spans of TAD-like domains are mostly conserved between tissues, though substantial additional work would be required to quantify the strength/prominence of these TAD-like domains sufficiently to compare them on a quantitative level. We share the reviewer’s desire for data capable of making such distinctions. This is a challenge that we intend to solve in future work with much more highly replicated chromatin conformation data, which we think is necessary to provide a rigorous quantification of domain state across tissues. Nevertheless, we are confident that, even with the data here, we can confidently put a lower bound on the level of conservation between tissues, though admittedly, we are only capturing a subset of their differences.

For example, of the domains (inferred by TopDom) found only in a single tissue, about 56.4-86.4% are found only in a single replicate whereas 13.6-43.6% (which corresponds to 0.6-3.5% of the total domains) are found in both replicates (they are more confident to be tissue-specific). Thus, we can only confidently conclude that as much as 0.6-3.5% of domains might be tissue-specific as they are supported by two replications. For those sample-specific domains, likely resulting from other variations, such as the batch effect, suggesting that more replications are needed for the rigorous annotation of tissue-specific domains. Similar results were obtained using TAD-like domains inferred from TADtool (Supplementary Note 3).

In the revised manuscript, we revise our interpretation “highly conserved” instead by noting that “Our results suggest that as much as 0.6-3.5% of domains might be limited to only one of the tissues investigated here. Future work with higher replication will permit rigorous annotation of

tissue-specific domains, allowing us to quantify the degree of divergence and conservation between tissues.” (see changes on Page 9-10)

Figure: TAD inferred by TADtool (see details in Supplementary Note 3)

Reviewer #3 (Remarks to the Author):

I am satisfied for the author's response for the reviewer's concerns. I agree that the current ms is enough for publication in this journal.

Thank you.

Reviewer #4 (Remarks to the Author):

The authors answered all my questions and incorporated all my comments in the final version of the manuscript. I do not see any further need for changes to publish this very nice work. Well done!

Thank you.

Reviewers' Comments:

Reviewer #2:

Remarks to the Author:

The authors have addressed all my concerns. Congratulations for this excellent work!

Final revision

Reviewer #2 (Remarks to the Author):

The authors have addressed all my concerns. Congratulations for this excellent work!

Thank you.